# Preference Learning Algorithms Do Not Learn Preference Rankings

**Angelica Chen**
New York University
ac5968@nyu.edu

**Sadhika Malladi**
Princeton University
smalladi@princeton.edu

**Lily H. Zhang**
New York University
lily.h.zhang@nyu.edu

**Xinyi Chen**
Google DeepMind; Princeton University
xinyic@google.com

**Qiuyi Zhang**
Google DeepMind
qiuyiz@google.com

**Rajesh Ranganath**
New York University
rajeshr@cims.nyu.edu

**Kyunghyun Cho**
New York University; Genentech; CIFAR LMB
kyunghyun.cho@nyu.edu

## Abstract

Preference learning algorithms (e.g., RLHF and DPO) are frequently used to steer LLMs to produce generations that are more preferred by humans, but our understanding of their inner workings is still limited. In this work, we study the conventional wisdom that preference learning trains models to assign higher likelihoods to more preferred outputs than less preferred outputs, measured via *ranking accuracy*. Surprisingly, we find that most state-of-the-art preference-tuned models achieve a ranking accuracy of less than $60\%$ on common preference datasets. We also derive the *idealized ranking accuracy* that a preference-tuned LLM would achieve if it optimized the DPO or RLHF objective perfectly. We demonstrate that existing models exhibit a significant *alignment gap – i.e.*, a gap between the observed and idealized ranking accuracies. We attribute this discrepancy to the DPO objective, which is empirically and theoretically ill-suited to fix even mild ranking errors in the reference model, and derive a simple and efficient formula for quantifying the difficulty of learning a given preference datapoint. Finally, we show that ranking accuracy strongly correlates with the empirically popular win rate metric when the model is close to the reference model, shedding further light on the differences between on-policy (e.g., RLHF) and off-policy (e.g., DPO) preference learning algorithms.

## 1   Introduction

Recent work on aligning LLMs has focused predominantly on tuning models to adhere to human preferences – commonly through reinforcement learning (RLHF; Stiennon et al. [47]) or directly via offline supervision (DPO; Rafailov et al. [41]). Preference learning algorithms [20, 55, 58] were originally designed to use a dataset of pairwise preferences over candidates to train a model with high *ranking accuracy* – that is, the model can precisely rank preferred outputs over dispreferred ones. In the case of language models, the ranking is determined by the likelihood assigned to each candidate.

Many LLM alignment techniques are designed to yield models with a high preference ranking accuracy, including SLiC [68, 67], RAFT [8], PRO [46], and RRHF [64]. Most prominently, Rafailov et al. [41] claimed that their popular direct preference optimization (DPO) algorithm "increases

38th Conference on Neural Information Processing Systems (NeurIPS 2024).

the relative log probability of preferred to dispreferred response." It is standard to evaluate these various objectives by measuring how often the resulting model's generations are preferred over another model's (i.e., a *win rate*) [71]. However, the relationship between the loss, ranking accuracy, and win rate is unclear, leaving open the question of what these alignment techniques are actually accomplishing during training.

In this work, we demonstrate that RLHF and DPO struggle to increase ranking accuracy in practice and explore both the theoretical and empirical reasons why. Our findings highlight an intricate relationship between offline optimization and online behavior, and motivate the need for more fine-grained analyses of preference training dynamics. Our contributions are as follows:

1. **Existing models do not achieve high ranking accuracies.** We demonstrate that a wide variety of open-access preference-tuned LLMs (e.g., LLAMA 2 7B CHAT, GEMMA 7B IT, and ZEPHYR 7B DPO) achieve a ranking accuracy below $60\%$ across a range of validation splits from commonly used preference datasets, such as UltraFeedback [7], Anthropic helpfulness and harmlessness (HH-RLHF, [14]), and Stanford Human Preferences (SHP, [11]) (Figure 1). Although we do not advocate for ranking accuracy as a measure of model quality, we analyze LLMs' ranking accuracies nonetheless because (1) ranking accuracy has motivated the design of many preference learning algorithms, and (2) DPO directly optimizes for ranking accuracy (Theorem 3.1).

2. **Existing models exhibit a significant *alignment gap* between the ranking accuracy they achieve and the accuracy achievable under idealized conditions.** We derive a simple formula (Theorem 3.1) for the *idealized ranking accuracy* (i.e., the ranking accuracy achieved from training on ground-truth preference data and perfectly optimizing the DPO or RLHF objective). We observe that models suffer from a significant *alignment gap* in that they achieve ranking accuracy far below the idealized ranking accuracy (Table 1, Figure 1).

3. **Preference learning rarely corrects incorrect rankings.** We prove theoretically that even mild ranking errors in the reference model can make it virtually impossible for DPO and its variants to correct the ranking (Theorem 4.1), and demonstrate that in practice, the rankings are rarely flipped (Fig. 2) and the reference model likelihoods generally determine the ranking accuracy (Fig. 3). Our results permit straightforward and efficient identification of hard-to-learn preference datapoints without any tuning.

4. **Ranking accuracy and win rate are closely correlated when the model is close to the reference model.** We observe that the ranking accuracy and win rate trend together when the model is close to the reference model during the early phase of alignment, but become anti-correlated once the model has moved too far away, adding to the ongoing discussion on the differences between on-policy and off-policy behaviors of preference-tuned LLMs.

Crucially, our work highlights fundamental flaws in RLHF and DPO that prevent the preference-tuned model from achieving a high ranking accuracy *even on the training dataset*.

## 2 Preliminaries

### 2.1 Learning from Human Preferences

**Preference Data**    Human preference data typically takes the form of pairwise preferences. Each prompt $x$ is paired with two possible continuations – $y_1$ and $y_2$. One or more human raters then annotate which continuation is preferred. When there are multiple raters, we use $\alpha(x, y_1, y_2)$ to denote the proportion of raters who prefer $y_1$ over $y_2$.[1]

**Definition 2.1** (Aggregated Preference Datapoint). Consider a prompt $x$ with two possible continuations $y_1$ and $y_2$ and the proportion of raters $\alpha(x, y_1, y_2)$ who preferred $y_1$ over $y_2$. Then, the aggregated preference datapoint for each prompt $x$ is denoted $(x, y_w, y_l)$ where $y_w$ is the completion preferred by the majority of voters.

We note that at the time of writing, the vast majority of datasets either use a single rater [14] or only release aggregated preference data [11, 26], so we often do not have access to $\alpha(x, y_1, y_2)$.

---

[1]In the limit, when there are infinite raters, the empirical proportion $\alpha(x, y_1, y_2)$ converges to the ground truth preference $\mathbb{P}[y_1 \succ y_2 \mid x]$.

A standard assumption is that the ground-truth human preferences obey the Bradley-Terry model (Assumption A.1).

**Supervised Fine-Tuning (SFT)**  In the first step of the preference learning pipeline, the model is typically trained using the standard cross-entropy objective on some choice of offline instruction-tuning dataset(s). In some implementations [52], a variety of third-party datasets are selected, whereas in other implementations [47, 41, 43] the model is instead trained on the preferred continuations $(x, y_w)$ from the same preference learning dataset that is used in downstream preference learning. The resulting model is often used as a *reference model*, denoted as $\pi_{\text{Ref}}$ or $\pi_{\text{SFT}}$, and it typically serves as the initialization when learning from human preferences.

**Reinforcement Learning from Human Feedback (RLHF)**  Learning from human feedback originally required using reinforcement learning [47]. In this setting, the possible continuations for each prompt are sampled from a reference model (i.e., $(y_w, y_l) \sim \pi_{\text{Ref}}(\cdot \mid x)$) and then annotated and aggregated to create a preference dataset $\mathcal{D}$. Then, one frames the problem as binary classification between the two continuations and trains a reward model $r_\phi(x, y)$ to minimize $\mathcal{L}_R(r_\phi, \mathcal{D}) = -\mathbb{E}_{(x,y_w,y_l) \sim \mathcal{D}}[\log \sigma(r_\phi(x, y_w) - r_\phi(x, y_l))]$. Finally, one trains the model $\pi_\theta$ to maximize the reward without straying too far from the reference model $\pi_{\text{Ref}}$. Because sampling generations from the model is non-differentiable, it is common to use PPO to maximize the reward $r(x, y) = r_\phi(x, y) - \beta(\log \pi_\theta(y \mid x) - \log \pi_{\text{Ref}}(y \mid x))$, where $\beta > 0$ is a regularization coefficient designed to prevent the model from straying too far from its initialization.

**Preference Learning with DPO**  Rafailov et al. [41] demonstrated that one can avoid using PPO by reparametrizing the objective to operate over policies instead of over rewards. Then, one can minimize the differentiable DPO objective.

**Definition 2.2** (DPO Objective [41])**.**  Let $\sigma$ be the sigmoid function and $\beta > 0$ be a hyperparameter. Then, the DPO objective for an aggregated preference dataset $\mathcal{D}$ and a reference model $\pi_{\text{Ref}}$ is defined as

$$\mathcal{L}_{\text{DPO}}(\pi_\theta, \pi_{\text{Ref}}) = - \mathop{\mathbb{E}}_{(x,y_w,y_l) \sim \mathcal{D}} \left[ \log \sigma \left( \underbrace{\beta \log \frac{\pi_\theta(y_w \mid x)}{\pi_{\text{Ref}}(y_w \mid x)} - \beta \log \frac{\pi_\theta(y_l \mid x)}{\pi_{\text{Ref}}(y_l \mid x)}}_{\text{reward margin}} \right) \right]$$

$$= - \mathop{\mathbb{E}}_{(x,y_w,y_l) \sim \mathcal{D}} \left[ \log \sigma \left( \underbrace{\beta \log \frac{\pi_\theta(y_w \mid x)}{\pi_\theta(y_l \mid x)}}_{\text{model log-ratio}} + \beta \underbrace{\log \frac{\pi_{\text{Ref}}(y_l \mid x)}{\pi_{\text{Ref}}(y_w \mid x)}}_{\text{reference model log-ratio}} \right) \right]$$

We denote the DPO loss on the aggregated datapoint $(x, y_w, y_l)$ as $\mathcal{L}_{\text{DPO}}(x, y_w, y_l; \pi_\theta, \pi_{\text{Ref}})$.

## 2.2  Evaluation Metrics

Evaluating the alignment of a preference-tuned LLM is both under-specified and multi-dimensional. Many knowledge-based and logic-based benchmarks (e.g. MMLU, GLUE, BIG-Bench, HELM) already exist, but these benchmarks largely fail to capture nuanced aspects of human preference, such as helpfulness or harmlessness [14]. As such, one standard evaluation is to ask human or machine raters how often the model produces a favorable completion compared to a baseline (*i.e.*, win rate). Human win rate is the gold standard but is costly to compute and can be biased based on size and nature of the worker pool [19, 25]. Rating completions using another LLM (e.g., MT-bench) can be cheaper but similarly suffers from various biases [39, 69, 62], and several studies have revealed failures in many LLM judges to identify violations of instruction-following [66, 29]. Nevertheless, since win rate evaluations are so prevalent, we compare ranking accuracy against win rate in Sec. 5 and describe when the former off-policy metric is correlated with the popular on-policy metric.

Besides the win rate, preference learning algorithms are also benchmarked by the frontier of the rewards versus the divergence from the initialization [41], which serves as a heuristic of how well the model can incorporate preference data without unlearning prior information. However, it is unclear how well rewards can describe the success of alignment.

As aforementioned, the current paper investigates the *ranking accuracy*, which is defined as follows:

**Definition 2.3** (Ranking Accuracy)**.** The ranking accuracy $\mathcal{R}$ of a model $\pi_\theta$ on an aggregated preference datapoint $(x, y_w, y_l)$ is defined as

$$\mathcal{R}(x, y_w, y_l; \pi_\theta) = \begin{cases} 1 & \pi_\theta(y_w \mid x) \geq \pi_\theta(y_l \mid x) \\ 0 & \text{otherwise.} \end{cases} \tag{1}$$

Analogously, the ranking accuracy of policy $\pi_\theta$ on a dataset $\mathcal{D} = \{(x, y_w, y_l)\}$ is $\mathcal{R}(\mathcal{D}; \pi_\theta) = \mathbb{E}_{(x, y_w, y_l) \sim \mathcal{D}} \mathcal{R}(x, y_w, y_l; \pi_\theta)$. In the rare case where a dataset has more than two outputs $y$ per prompt $x$, we use the generalized ranking accuracy definition stated in App. A.6. We do not advocate for ranking accuracy as a metric of model quality, but as a lens into the inner workings of common preference learning algorithms.

**Remark 2.4** (Lengths of Completions)**.** We note that $y_w$ and $y_l$ can have different lengths; for example, Singhal et al. [45] showed that $y_w$ is usually longer. Length can deflate $\pi_\theta(y_w \mid x)$ and reduce the ranking accuracy. One can normalize the likelihoods by the length of the response, but the length-normalized ranking accuracy may not be meaningful in practice, because it is currently unclear how to sample from the length-normalized likelihood. For completeness, we report the ranking accuracies of both the unnormalized and normalized policies, denoted $\mathcal{R}$ and $\tilde{\mathcal{R}}$, respectively.

**Remark 2.5** (Difference between Ranking Accuracy and Reward Accuracy)**.** For RLHF models and DPO models, the ranking accuracy is not equivalent to the reward accuracy (*i.e.*, the metrics evaluated in RewardBench [29]). In the RLHF case, we are evaluating the ranking accuracy of the final policy rather than the reward model. In the DPO case, reward accuracy measures whether $\beta \log \frac{\pi_\theta(y_w \mid x)}{\pi_{\text{Ref}}(y_w \mid x)} > \beta \log \frac{\pi_\theta(y_l \mid x)}{\pi_{\text{Ref}}(y_l \mid x)}$ instead of whether $\pi_\theta(y_w \mid x) > \pi_\theta(y_l \mid x)$. Since we ultimately sample from $\pi_\theta$ rather than $\frac{\pi_\theta(y \mid x)}{\pi_{\text{Ref}}(y \mid x)}$, we find the ranking accuracy to be of greater practical importance.

Moreover, we demonstrate that under very stringent conditions, minimizing the DPO objective results in a model with high ranking accuracy. We characterize the phenomenon on individual datapoints, as is the case throughout the paper, but note that Markov's inequality can be straightforwardly applied to extend the results to a low loss on the entire dataset.

**Proposition 2.6** (Sanity Check)**.** *Recall the definition of $y_w$, $y_l$ in Definition 2.1. If $\pi_{Ref}(y_w \mid x) \geq \pi_{Ref}(y_l \mid x)$ and $\mathcal{L}_{DPO}(x, y_w, y_l; \pi_\theta, \pi_{Ref}) \leq 0.6$, then $\mathcal{R}(x, y_w, y_l) = 1$.*

This result, proved in App. A.1, requires the condition that the reference model already has the correct ranking, so it is unlikely to hold across all datapoints in practice and somewhat moot. The remainder of the paper focuses on more realistic settings where the reference model is imperfect.

## 3   The Alignment Gap

Prop. 2.6 showed that training a low DPO loss with a perfect reference model yields a model with perfect ranking accuracy. However, Fig. 1a shows that real-world reference models exhibit low ranking accuracies, which prompts us to study more realistic, imperfect reference models.

### 3.1   Existing Reference Models Rarely Have Correct Rankings

Fig. 1a indicates that reference models rarely achieve high ranking accuracy on common preference datasets (except Synthetic Instruct GPT-J Pairwise), even though many were likely trained on the preferred completions (see Sef. 2.1). Many of the models do not have documented training data so we do not know which preference datasets, if any, are in-distribution. We also fine-tune several pretrained LLMs on the preferred completions (see App. B.1) and observe that ranking accuracy does not increase significantly.[2] Based on our findings, we turn to the case of imperfect reference models.

### 3.2   Idealized Ranking Accuracy

We showed above that empirically, reference models exhibit poor accuracy when ranking the plausible completions. However, the RLHF reward and DPO objective were formulated to ensure that the

---

[2]It is not surprising that fine-tuning on the preferred completions does not boost ranking accuracy, since the model does not receive any knowledge of the relative qualities of the preferred and rejected completions.

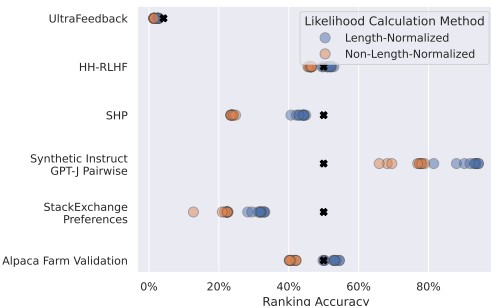 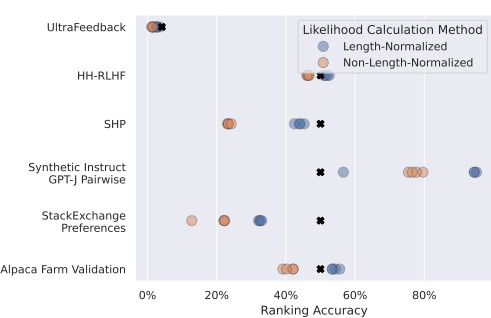

(a) Ranking accuracies of various reference models, including GPT2 [40], PYTHIA 2.8B [4], PYTHIA 1.4B [4], LLAMA 2 7B [52], VICUNA 1.5 7B [69], OLMO 7B [17], TULU2 7B [21], ZEPHYR 7B SFT [54], MISTRAL V0.1 7B [23], and GEMMA 7B [51]

(b) Ranking accuracies of various preference-tuned models, including LLAMA 2 7B CHAT [52], TULU2 7B DPO [21], ZEPHYR 7B DPO [54], and GEMMA 7B IT [51]

Figure 1: **Both reference and preference-tuned models exhibit low ranking accuracy on most preference datasets.** Each point represents the length-normalized or non-length-normalized ranking accuracy of individual (1a) reference models (pre-trained or fine-tuned), or (1b) preference-tuned models (trained with DPO or RLHF). The random chance accuracy for each dataset is indicated with a black 'X'. We sub-sample 1K examples from each dataset and use the test split when available. We describe datasets in B.2 and list all numbers in Tables 2, 3, and 4. For UltraFeedback, ranking accuracy is measured with exact match across all 4 outputs (see App. A.6).

model learns the preference dataset but does not move too far from the reference model $\pi_{Ref}$, so there may be a limit on the possible accuracy of the preference-tuned model. Here, we formalize this intuition by studying what the optimal policies would be when perfectly optimizing DPO or RLHF with access to perfect data (i.e., true proportions of human preferences).[3]

**Theorem 3.1** (Simulating Perfect RLHF[4]). *Fix a reference model $\pi_{Ref}$ and an aggregated preference datapoint $(x, y_w, y_l) \sim \mathcal{D}$. Assume the dataset includes the ground-truth human preferences: that is, $\alpha(x, y_w, y_l) = \mathbb{P}(y_w \succ y_l)$, and that these preferences obey the Bradley-Terry model (Assumption A.1). Let $\pi^*$ be the model resulting from perfectly optimizing the DPO or RLHF objective on $(x, y_w, y_l)$ as described in Section 2.1. Then, $\pi^*$ satisfies*

$$\frac{\pi^*(y_w \mid x)}{\pi^*(y_l \mid x)} = \frac{\pi_{Ref}(y_w \mid x)}{\pi_{Ref}(y_l \mid x)} \left( \frac{\alpha(x, y_w, y_l)}{1 - \alpha(x, y_w, y_l)} \right)^{1/\beta} \tag{2}$$

*where $\alpha(x, y_w, y_l)$ is the proportion of raters who preferred $y_w$ over $y_l$ and $\beta$ is a hyperparameter in the DPO and RLHF objectives.*

**Remark 3.2.** We prove this result in App. A.2. Note that deterministic preferences, i.e., $\alpha(x, y_w, y_l) = 1$, should not be confused with settings that only report a single individual's preference, i.e., $\hat{\alpha}(x, y_w, y_l) = 1$. In the former, the optimal probability ratio is infinity. The latter requires a better estimate of $\mathbb{P}(y_w \succ y_l)$, e.g., more samples.

This result allows us to simulate the policy resulting from perfect optimization of either the RLHF or the DPO learning objective. As such, given a reference model $\pi_{Ref}$ and preference dataset $\mathcal{D}$, we can easily measure the *idealized ranking accuracy* of a model. We prove this result in App. A.3.

**Corollary 3.3** (Idealized Ranking Accuracy). *Given a reference model $\pi_{Ref}$, the DPO or RLHF hyperparameter $\beta$, a dataset of aggregated preferences $\mathcal{D} = \{(x, y_w, y_l)\}$ and their corresponding rater proportions $\alpha(x, y_w, y_l)$, the ranking accuracy of the optimum of the RLHF or DPO objective $\pi^*$ is given by*

$$\mathcal{R}^*(\mathcal{D}; \pi_{Ref}) = \mathop{\mathbb{E}}_{(x, y_w, y_l) \sim \mathcal{D}} \left[ \mathbb{1} \left[ \frac{\pi_{Ref}(y_w \mid x)}{\pi_{Ref}(y_l \mid x)} \left( \frac{\alpha(x, y_w, y_l)}{1 - \alpha(x, y_w, y_l)} \right)^{1/\beta} > 1 \right] \right] \tag{3}$$

---

[3]This result differs from Proposition 2.6 in that it accounts for a potentially imperfect reference model.
[4]We note that this result can also be straightforwardly derived from prior works [38, 27, 16].

Table 1: **The idealized ranking accuracy of existing algorithms is not perfect, but preference-tuned models exhibit ranking accuracies far even from this idealized case.** We provide both the length-normalized ($\tilde{\mathcal{R}}$) and non-length-normalized ($\mathcal{R}$) ranking accuracies for a variety of open-access preference-tuned models on the Alpaca Farm [9] validation dataset (described in App. B.2). We also provide the idealized ranking accuracy ($\mathcal{R}^*$ or $\tilde{\mathcal{R}}^*$, Corollary 3.3). Since idealized ranking accuracy can be computed with a variety of values of $\beta$, we provide the minimum, median, and maximum idealized ranking accuracy values for a range of $\beta$. For more details, see App. B.4.

| Preference-Tuned Model | Length-Normalized | | Non-Length-Normalized | |
|---|---|---|---|---|
| | $\tilde{\mathcal{R}}$ | $\tilde{\mathcal{R}}^*$ (Min./Med./Max.) | $\mathcal{R}$ | $\mathcal{R}^*$ (Min./Med./Max.) |
| ZEPHYR-7B-DPO | 54% | 86% / 98% / 100% | 42% | 90% / 99% / 100% |
| TULU-2-DPO-7B | 53% | 87% / 97% / 100% | 42% | 91% / 99% / 100% |
| GOOGLE-GEMMA-7B-IT | 54% | 73% / 73% / 97% | 40% | 67% / 93% / 100% |
| LLAMA-2-7B-CHAT-HF | 53% | 87% / 97% / 100% | 40% | 91% / 99% / 100% |

where $\mathbb{1}[\cdot]$ *is the indicator function. When computed on length-normalized likelihoods from* $\pi_{\tilde{Ref}}$*, we denote the idealized ranking accuracy as* $\tilde{\mathcal{R}}^*$*.*

## 3.3 Measuring the Alignment Gap

Given access to $\pi_{\text{Ref}}$, $\beta$, and the $\alpha(x, y_w, y_l)$ values for each triple $(x, y_w, y_l)$ in a given preference dataset, we can compute the idealized ranking accuracy from Eq. 3.[5] The results are shown in Table 1 and further details are given in App. B.4.

We identify several surprising findings. Firstly, even under ideal conditions (*i.e.* perfectly optimizing the objective on ground-truth preference data), the idealized ranking accuracy is still sometimes below 100%. This distance varies with the choice of $\beta$, which indicates that the limits of DPO/RLHF depend largely upon how strong the reliance on $\pi_{\text{Ref}}$ is. Furthermore, we find that many state-of-the-art models do not achieve a ranking accuracy anywhere close to the idealized ranking accuracy, exhibiting alignment gaps ranging from 19 to 59 percentage points (measured to the median idealized $\mathcal{R}$ or $\tilde{\mathcal{R}}$).

## 4 Understanding Ranking Accuracy with DPO

We now turn to the training objectives to account for the alignment gap. We focus our analysis on the DPO objective (Definition 2.2), because its failure to achieve high ranking accuracy is especially surprising (Table 1). In particular, DPO directly maximizes the reward margin between preferred-dispreferred pairs over an offline dataset so we would expect it to perform well on in-distribution held-out data. We also note that DPO is a popular choice in the community for aligning LLMs, because it is less costly than performing RLHF.

In this section, we study real-world characteristics of DPO. First, we show empirically that DPO rarely flips the ranking of the two continuations. This result combined with the observation that reference models exhibit poor ranking accuracy (Sec. 3.1) provides an explanation for the observed poor ranking accuracies in Table 1. We then formally characterize how hard it is for DPO to correct the ranking of each datapoint. Our result highlights how the reference model conditions the optimization: as the reference model log-ratio (Definition 2.2) grows larger, one has to reduce the DPO loss to a dramatically small value to flip the incorrect ranking (Fig. 3).

### 4.1 DPO Rarely Flips Preference Rankings

To study how ranking accuracy changes over the course of DPO training, we train three sizes of models (GPT-2 [40], Pythia 2.8B [4], and Llama 2-7B [52]) across three seeds each on the Anthropic HH-RLHF [3] preference dataset and study the ranking accuracy on different partitions of the training dataset. We present the results from training one seed of Pythia 2.8B in Fig. 2, and defer training

---

[5]Note that when $\alpha(x, y_w, y_l) = 1$, we replace it with $1 - \epsilon$ to compute the formula. See App. B.4.

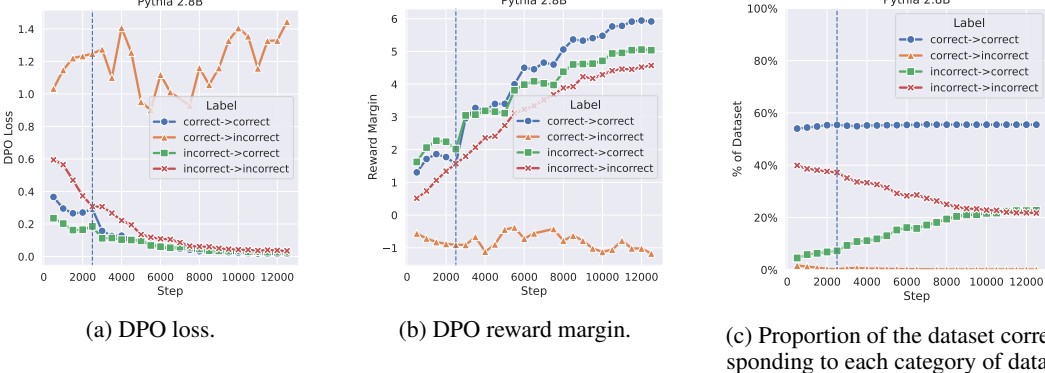

(a) DPO loss.

(b) DPO reward margin.

(c) Proportion of the dataset corresponding to each category of data.

Figure 2: **Despite continuously decreasing the loss, DPO rarely flips the rankings of pairs before the point of overfitting (marked by the vertical dashed line) and instead mostly increases the reward margin of already correctly ranked pairs.** We train a Pythia-2.8B model for 5 epochs using the DPO objective and categorize the training dataset into four subsets – examples that initially have the correct ranking and are flipped to (1) correct or (2) incorrect, and examples that initially have the incorrect ranking and are flipped to (3) correct or (4) incorrect. In all three figures, the hue of the point indicates the category. The dashed vertical line indicates the training step at which the lowest eval. loss occurs. Past this point, the model begins to overfit (*i.e.*, the eval. loss starts to increase). We also present results for two other models with three seeds each in Appendix C.

details to App. C.1 and results on the other two models to App. C.2. In Fig. 2, we partition a random subsample of 1K examples from the training dataset into four groups based on whether the reference model $\pi_{\text{Ref}}$ had the correct ranking and whether the current model $\pi_\theta$ has the correct ranking.

Surprisingly, Fig. 2 demonstrates that DPO rarely flips the ranking of $(y_w, y_l)$ over the course of training despite consistently reducing the loss $\mathcal{L}_{\text{DPO}}$. Aside from the group of points for which the model unlearns the correct preference ranking, we observe that the loss decreases and the reward margin increases consistently while training. However, **at the point of lowest validation loss (marked by the vertical dashed line in Fig. 2c), less than $10\%$ of the originally incorrectly ranked points have been flipped to have the correct ranking**. Past this point, the model begins to overfit (*i.e.*, the validation loss begins to increase). DPO does not substantially improve the ranking accuracy until far past the point of overfitting. Although Theorem 3.1 predicts that the theoretically optimal DPO model exhibits high ranking accuracy, in practice the empirical endpoint of training occurs far from the theoretical optimum. This indicates that the DPO objective is ill-formulated to induce a high ranking accuracy in practice.

## 4.2 Analysis: How Easy Is It To Flip A Ranking?

In the result below, we show that the DPO loss can decrease substantially without any improvement on the ranking accuracy of the model. Specifically, the DPO loss that the model needs to reach in order to have the correct ranking on an example $(x, y_w, y_l)$ depends on the quality of the reference model, quantified by the reference model log-ratio. This dependence is highly ill-conditioned, whereby using a reference model with moderately incorrect likelihoods assigned to each continuation can effectively prevent DPO from learning the correct ranking.

**Theorem 4.1.** *Consider an aggregated preference datapoint $(x, y_w, y_l)$ such that the reference model log-ratio is some constant $c$, i.e. $\log \frac{\pi_{Ref}(y_l|x)}{\pi_{Ref}(y_w|x)} = c$. Then, $\mathcal{R}(x, y_w, y_l) = 1$ if and only if $\mathcal{L}_{DPO}(x, y_w, y_l) \leq -\log \sigma(\beta c)$, where $\sigma$ is the sigmoid function.*

**Remark 4.2.** It is straightforward to extend our analysis to popular variants of DPO. For illustration, we prove an analogous result for identity preference optimization (IPO, Azar et al. [2]) in App. A.5.

We prove this result in App. A.4. Our theoretical result allows us to formally identify the points that will be hard to flip in their rankings. Fig. 3 visualizes the reference model log-ratio for several settings and highlights that datapoints with even mild ranking errors in the reference model will require the loss to be reduced to a very low value in order to flip the ranking. App. E contains examples of hard-

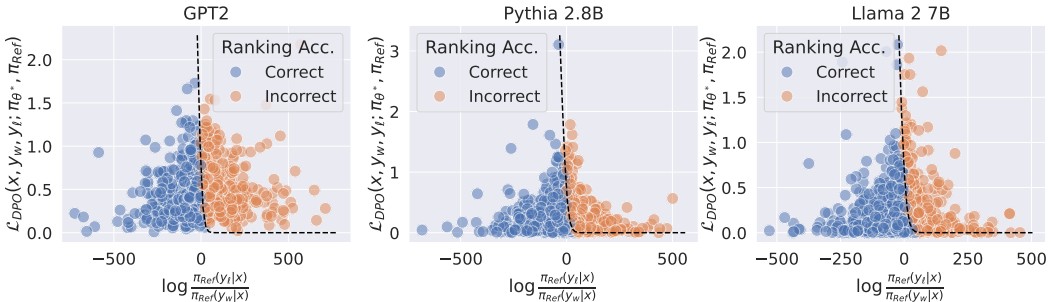

Figure 3: **DPO loss alone does not predict ranking accuracy, due to the influence of the reference model log-ratio in the loss.** Each point represents the DPO loss on a separate training example $(x, y_w, y_l)$ from a subsample of 1K examples from the training dataset, using the model $\pi_{\theta*}$ that corresponds to the checkpoint with the lowest validation loss. The color of each point indicates whether $\pi_{\theta*}$ achieves the correct ranking on that example, *i.e.*, whether $\pi_{\theta*}(y_w|x) > \pi_{\theta*}(y_l|x)$. The dashed line is the function $f(c) = -\log \sigma(\beta c)$, from Theorem 4.1. In summary, the examples that $\pi_{\theta*}$ classifies correctly tend to be those that were already classified correctly by the reference model. Results for the other two seeds of each model are given in Fig. 8.

to-learn, easy-to-learn, and easy-to-flip datapoints. We observe that the hard-to-learn datapoints are substantially longer than the easy ones, and that the easy datapoints generally contain less ambiguous preference annotations. More generally, our result motivates the use of stronger $\pi_{\text{Ref}}$ models and iterative or on-policy variants of DPO [50, 65, 24, 53].

## 5 Ranking Accuracy and Win Rate

Our results on ranking accuracy illuminate how well DPO and RLHF can align to preference data, but we have not yet related these insights to how the generative behavior of the model changes during alignment. In particular, ranking accuracy is a convenient but off-policy metric and is thus not as widely adopted as the on-policy metric of win rate (see Sec. 2.2). Indeed, one could maximize the ranking accuracy by learning a strong classifier on the preference data, but that model may not generate high-quality text. Here, we explore the gap between on-policy (i.e., generative) and off-policy (i.e., classification) behaviors of LLMs through the lens of ranking accuracy and win rate. Since the DPO objective directly optimizes for ranking accuracy (Proposition 2.6), the relationship between these two metrics is a direct reflection of how off-policy training affects on-policy behavior.

We study the relationship between win rate and ranking accuracy in two settings: (1) during DPO training, and (2) in a DPO variant modulating the influence of $\pi_{\text{Ref}}$. We measure the win rate on 500 responses to prompts from the training dataset using the Alpaca Eval GPT-4 [30] auto-annotator.

**Setting 1: DPO Training.** We measure the win rate and the ranking accuracy of a Pythia 2.8B model [4] during DPO training with the same configuration as in Section 4. See Fig. 9 for the results.

**Setting 2: Attenuating the reference model.** Theorem 4.1 showed that $\pi_{\text{Ref}}$ exerts a negative influence on the ranking accuracy in most cases, so we design a new objective that scales the reference model log-ratio in $\mathcal{L}_{\text{DPO}}$ to further characterize how win rate and ranking accuracy relate.

$$\mathcal{L}_{DPO}^{\gamma}(\pi_\theta, \pi_{\text{Ref}}) = - \mathbb{E}_{(x, y_w, y_l) \sim \mathcal{D}} \left[ \log \sigma \left( \beta \left( \log \frac{\pi_\theta(y_w \mid x)}{\pi_\theta(y_l \mid x)} + \gamma \log \frac{\pi_{\text{Ref}}(y_l \mid x)}{\pi_{\text{Ref}}(y_w \mid x)} \right) \right) \right] \quad (4)$$

Note that $\mathcal{L}_{\text{DPO}}^{\gamma} = \mathcal{L}_{\text{DPO}}$ (Definition 2.2) when $\gamma = 1$, and a larger value of $\gamma$ increases the role of the reference model. Also, $\gamma$ directly scales $c$ in Theorem 4.1, thereby controlling how easy it is to fit the data and increase the ranking accuracy. We train a range of Pythia-2.8B models using the $\mathcal{L}_{DPO}^{\gamma}$ objective for $\gamma \in \{0, 0.25, 0.5, 0.75, 1.0, 1.25, 1.5, 1.75, 2.0\}$ and measure the ranking accuracies and win rates of the best model for each $\gamma$ value.[6]

---

[6]We use the best hyperparameters obtained from the experiments in Sec. 4.

**Takeaway: Ranking accuracy correlates with win rate when the model is close to the reference model.** In both settings, we observe that the win rate and ranking accuracy are highly correlated with one another in the early phase of training but become anti-correlated (i.e., ranking accuracy increases but win rate declines) as the model $\pi_\theta$ moves away from the reference $\pi_{\text{Ref}}$ (Fig. 4). Unlike traditional overfitting, the test loss is continuing to decline at this point (Fig. 9b). Experiments in Fig. 10 with the attenuated objective in Equation (4) further show that ranking accuracy and win rate trend together when the influence of the reference model is stronger (i.e., $\gamma$ is larger).

We speculate that when the model is far from the reference model, overly optimizing the reward margin can harm the generative capabilities of the model, which are primarily acquired during pre-training. In other words, the off-policy behavior of the model can no longer predictably describe the on-policy generations when the reference model used in the offline objective is far from the current model. Our findings confirm the fundamental tradeoff between fitting the preference data and maintaining generative capabilities acquired during pre-training [22] and align with prior observations that adding on-policy preference data can make offline learning more effective [50, 65, 24, 53].

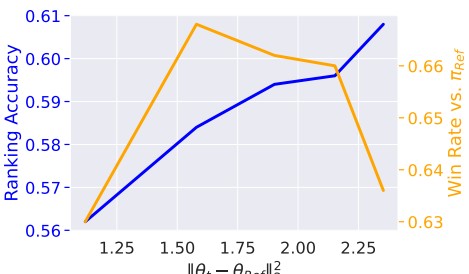

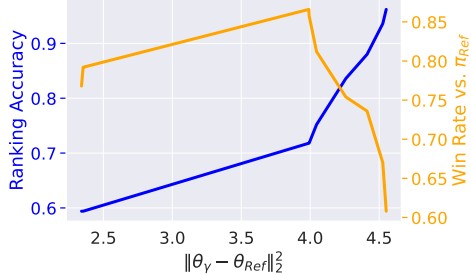

(a) Ranking accuracy and win rate of various Pythia 2.8B checkpoints during a single epoch of DPO training, versus the distance travelled by the model weights $\theta_t$ from the initialization.

(b) Ranking accuracy and win rate of various $\gamma$-scaled models (trained with $\mathcal{L}_{\text{DPO}}^\gamma$ for 5 epochs), versus the distance travelled by the model weights $\theta_\gamma$ from the initialization.

Figure 4: **When the model weights have not travelled far from $\theta_{\text{Ref}}$, ranking accuracy and win rate increase together.** $\theta_t$ represents the model weights at checkpoint $t$ during DPO training, and $\theta_\gamma$ represents the weights for a model trained to convergence with $\mathcal{L}_{\text{DPO}}^\gamma$.

## 6 Related Work

**Analyses of Preference Learning Algorithms** Many works have investigated the role of the preference dataset [56, 61], the reliability of the evaluations [69, 29], and the confounding factor of response length [45, 10, 57, 37]. Theoretical works have unified the many preference learning algorithms into clear taxonomies that permit analysis and, sometimes, yield new variants [2, 61, 48, 50, 63, 34]. Several works study idiosyncrasies of preference learning, such as why DPO decreases the likelihood of both rejected and chosen outputs from the dataset [42, 13, 35] and why RLHF exhibits vanishing gradients [44]. In contrast, our work approaches understanding DPO and RLHF through the lens of ranking accuracy, and our findings emphasize the role of the reference model regularization in preference learning. Relatedly, SliC-HF [67], CPO [59], and pairwise cringe loss [60] optimize log probability margins $\log \pi(y_w|x) - \log \pi(y_l|x)$, effectively removing the regularization toward the reference model. Liu et al. [32] recommend using the reference model at inference time to exert more granular control over the regularization. Meng et al. [34] remove the $\pi_{\text{Ref}}$ terms altogether and use a target reward margin to prevent over-optimization. Additionally, Chennakesavalu et al. [6] design a DPO-like objective that includes an additional hyperparameter controlling the strength of the $\pi_{\text{Ref}}$ terms, similar to our $\mathcal{L}_{DPO}^\gamma$ objective (Eq. 4). Tang et al. [50] also analyze the role of regularizing toward a reference model, though our work focuses the effect of this regularization on ranking accuracy.

**On-policy and Off-policy Preference Learning** Preference-tuning LLMs originally required using an on-policy algorithm [47], but many recent works have derived off-policy methods that can use a static preference dataset for supervision [41, 12, 18, 37]. Off-policy methods are preferred for their efficiency and ease of implementation, but several works have suggested that on-policy methods

are superior [48, 61, 49, 31]. Several iterative training methods aim to bridge this gap, where the reference model and the dataset are refreshed during the alignment procedure to contain annotated preferences on generations from the model at that point in training [65, 24, 53]. These intuitions align strongly with our observation that win rate and ranking accuracy, and thus, on-policy and off-policy behavior, are strongly correlated when the model is close to the reference model.

## 7 Discussion

Our work highlights the significant but nuanced relationship between preference learning and ranking accuracy. We have demonstrated both theoretically and empirically that RLHF and DPO struggle to teach the model to correctly rank preferred and dispreferred outputs, even in the training dataset. Although the learning objective promotes high ranking accuracy in theory (Proposition 2.6), we observed a prominent *alignment gap* resulting from the poor conditioning of reference models. We then drew connections between the off-policy nature of ranking accuracy and the on-policy evaluations of win rate, identifying specific scenarios in which on-policy behavior can or cannot be reliably predicted by off-policy behavior. App. 8 details the limitations of our work.

**Connections to Safety** Our work shows that it is difficult to steer pre-trained LLMs to adhere to even the preference data used for training. When LLMs are used to judge responses from other models [30, 69] or to improve their own abilities [33, 65], poor ranking accuracies can induce strong negative feedback loops that are costly to mitigate.

We also observe that win rate does not monotonically increase during training (Fig. 9a), despite the decrease in both train and test loss (Fig. 9b) and the modest gain in ranking accuracy (Fig. 9a). As such, it is clear that we still do not understand the behaviors of preference learning. For example, others have observed that DPO can cause the likelihoods of both chosen and rejected outputs to decrease [42, 13, 35, 36], which implies that the policy must be moving probability mass to possibly undesirable sequences outside the data distribution. Moreover, our investigation of the non-monotonic relationship between ranking accuracy and win rate emphasizes the need for concrete evaluations that can more reliably and transparently measure the success of preference learning.

**Future Work** Our theoretical results only describe the behavior of the model on the preference data used during training, but they can serve as a starting point for understanding generalization to different distributions of data, especially the one prescribed by the model itself [48]. Furthermore, we hope to analyze the optimization dynamics of preference learning, given the intriguing relationship observed between ranking accuracy and win rate. For instance, identifying when the win rate begins to diverge from the ranking accuracy can motivate adding fresh on-policy training data. Our initial investigation into ranking accuracy also suggests that it is worthwhile to explore how alignment techniques interact with other calibration metrics.

## 8 Limitations

Although we reproduce our main results (Sec. 4) on three types of models across three seeds each, these models are trained on a single dataset due to the computational constraints. Our theoretical results lead us to believe that our findings would generalize to other datasets, but empirical verification is still valuable. As mentioned previously, our work is optimization-agnostic and only describes model behavior on the training dataset, making it difficult to draw general claims about other distributions. In particular, we cannot rigorously describe the generative capabilities, though Sec. 5 initiates an investigation into when off-policy behavior can describe on-policy behaviors.

# 9 Acknowledgements

We thank Sanjeev Arora, Tianyu Gao, Eric Mitchell, Richard Pang, and Mengzhou Xia for helpful discussions during the development of this project. We also thank Nikita Nangia, Eshaan Nichani, and Alexander Wettig for their help in proofreading the work. This work was supported by National Science Foundation Award 1922658, the Samsung Advanced Institute of Technology (under the project Next Generation Deep Learning: From Pattern Recognition to AI), NIH/NHLBI Award R01HL148248, NSF CAREER Award 2145542, ONR N00014-23-1-2634, Apple, and Google. SM is additionally supported by ONR, NSF, and DARPA. We also thank Princeton Language and Intelligence (PLI) for computing resources and OpenAI credits, and NYU High Performance Computing (HPC) for computing resources and in-kind support.

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

# A Proofs and Additional Results

**Assumption A.1** (Bradley-Terry [5])**.** Given a prompt $x$ and two possible continuations $y_1$ and $y_2$, the ground truth human preference distribution satisfies

$$\mathbb{P}(y_1 \succ y_2 \mid x) = \frac{\exp(r^*(x, y_1))}{\exp(r^*(x, y_1)) + \exp(r^*(x, y_2))} \tag{5}$$

for some ground truth reward model $r^*$.

## A.1 Proof of Proposition 2.6

**Proposition A.2** (Reproduced from Proposition 2.6)**.** *Recall the definition of $y_w$, $y_l$ in Definition 2.1. If $\pi_{Ref}(y_w \mid x) \geq \pi_{Ref}(y_l \mid x)$ and $\mathcal{L}_{DPO}(x, y_w, y_l; \pi_\theta, \pi_{Ref}) \leq 0.6$, then $\mathcal{R}(x, y_w, y_l) = 1$.*

*Proof.* For notational convenience, write the log probability ratios as

$$a_x = \log \frac{\pi_\theta(y_w|x)}{\pi_{\text{Ref}}(y_w|x)}, \quad b_x = \log \frac{\pi_\theta(y_l|x)}{\pi_{\text{Ref}}(y_l|x)}.$$

Then we can express the probability that the model places on each response as

$$\pi_\theta(y_w|x) = \pi_{\text{Ref}}(y_w|x)e^{a_x}, \quad \pi_\theta(y_l|x) = \pi_{\text{Ref}}(y_l|x)e^{b_x}.$$

Suppose for $(x, y_w, y_l)$, $\mathcal{L}_{\text{DPO}}(x, y_w, y_l; \pi_\theta, \pi_{\text{Ref}}) \leq 0.6$. Expanding the DPO loss, we have

$$\mathcal{L}_{\text{DPO}}(x, y_w, y_l; \pi_\theta, \pi_{\text{Ref}}) = -\log \sigma(\beta(a_x - b_x)) \leq 0.6.$$

Rearranging the inequality and exponentiate on both sides, we have

$$\sigma(\beta(a_x - b_x)) \geq \exp(-0.6).$$

Note that $\sigma(0) = 0.5 < \exp(-0.6) \leq \sigma(\beta(a_x - b_x))$. Since the logistic function is monotonic, this implies that $0 \leq \beta(a_x - b_x)$ and $a_x \geq b_x$.

Writing out this last inequality,

$$\frac{\pi_\theta(y_w|x)}{\pi_{\text{Ref}}(y_w|x)} \geq \frac{\pi_\theta(y_l|x)}{\pi_{\text{Ref}}(y_l|x)} \Rightarrow \frac{\pi_\theta(y_w|x)}{\pi_\theta(y_l|x)} \geq \frac{\pi_{\text{Ref}}(y_w|x)}{\pi_{\text{Ref}}(y_l|x)} \geq 1,$$

where the last inequality is by the assumption on $\pi_{\text{Ref}}$. We conclude that $\mathcal{R}(x, y_w, y_l) = 1$ by definition. □

## A.2 Proof of Theorem 3.1

**Theorem A.3** (Simulating Perfect RLHF)**.** *Fix a reference model $\pi_{Ref}$ and an aggregated preference datapoint $(x, y_w, y_l) \sim \mathcal{D}$. Assume the dataset includes the ground-truth human preferences: that is, $\alpha(x, y_w, y_l) = \mathbb{P}(y_w \succ y_l)$, and that these preferences obey the Bradley-Terry model (Assumption A.1). Let $\pi^*$ be the (adequately expressive) model that perfectly minimizes the DPO objective on $(x, y_w, y_l)$, or perfectly maximizes the PPO objective on the optimal reward function as described in Section 2.1. Then, the optimal policy $\pi^*$ satisfies*

$$\frac{\pi^*(y_w \mid x)}{\pi^*(y_l \mid x)} = \frac{\pi_{Ref}(y_w \mid x)}{\pi_{Ref}(y_l \mid x)} \left( \frac{\alpha(x, y_w, y_l)}{1 - \alpha(x, y_w, y_l)} \right)^{1/\beta} \tag{6}$$

*where $\beta$ is a hyperparameter in the DPO and RLHF objectives.*

*Proof.* We first prove the statement for DPO. Following the notation from [41], fix a reward function $r$, let $\pi_r(y|x)$ be the optimal model under the KL-constrained RL objective given the reward function $r$. We can express $r(x, y)$ in terms of $\pi_r(y|x)$ and $\pi_{ref}(y|x)$:

$$r(x, y) = \beta \log \frac{\pi_r(y|x)}{\pi_{ref}(y|x)} + \beta \log Z(x),$$

where $Z(x)$ is the partition function for prompt $x$. Then, under the Bradley-Terry model, the probability of preferring $y_w$ over $y_l$ under the model $\pi_r$ is

$$\pi_r(y_w \succ y_l|x) = \frac{1}{1 + \exp(r(x,y_l) - r(x,y_w))}$$

$$= \frac{1}{1 + \exp(\beta \log \frac{\pi_r(y_l|x)}{\pi_{ref}(y_l|x)} - \beta \log \frac{\pi_r(y_w|x)}{\pi_{ref}(y_w|x)})}.$$

Given the ground-truth human preferences, DPO's maximum likelihood objective minimizes the binary classification loss on $(x, y_w, y_l)$:

$$\min_\pi \alpha(x, y_w, y_l) \log \pi(y_w \succ y_l|x) + (1 - \alpha(x, y_w, y_l)) \log(1 - \pi(y_w \succ y_l|x)).$$

Let $\pi^*$ denote an optimal policy from the loss above, the optimal preference probabilities satisfy

$$\pi^*(y_w \succ y_l|x) = \alpha(x, y_w, y_l),$$

and the optimal policy in turn satisfies

$$\frac{1}{1 + \exp(\beta \log \frac{\pi^*(y_l|x)}{\pi_{ref}(y_l|x)} - \beta \log \frac{\pi^*(y_w|x)}{\pi_{ref}(y_w|x)})} = \alpha(x, y_w, y_l).$$

Rearranging, we have

$$\frac{\alpha(x, y_l, y_w)}{\alpha(x, y_w, y_l)} = \exp\left(\beta \log \frac{\pi^*(y_l|x)}{\pi_{ref}(y_l|x)} - \beta \log \frac{\pi^*(y_w|x)}{\pi_{ref}(y_w|x)}\right),$$

taking a log on both sides and divide by $\beta$,

$$\frac{1}{\beta} \log \frac{\alpha(x, y_l, y_w)}{\alpha(x, y_w, y_l)} = \log \frac{\pi^*(y_l|x)\pi_{ref}(y_w|x)}{\pi_{ref}(y_l|x)\pi^*(y_w|x)},$$

and finally exponentiating both sides,

$$\frac{\pi^*(y_l|x)}{\pi^*(y_w|x)} = \frac{\pi_{ref}(y_l|x)}{\pi_{ref}(y_w|x)} \left(\frac{\alpha(x, y_l, y_w)}{\alpha(x, y_w, y_l)}\right)^{1/\beta},$$

the result follows by taking an inverse.

Now we show the result for RLHF, starting from the optimal solution of the KL-constrained reward maximization problem under the reward $r_\phi$, as derived in [41]:

$$\pi^*(y|x) \propto \pi_{\text{Ref}}(y|x) \exp\left(\frac{1}{\beta} r_\phi(x, y)\right).$$

The condition is straightforward to derive

$$\frac{\pi^*(y_w|x)}{\pi^*(y_l|x)} = \frac{\pi_{\text{Ref}}(y_w|x) \exp\left(\frac{1}{\beta} r_\phi(x, y_w)\right)}{\pi_{\text{Ref}}(y_l \mid x) \exp\left(\frac{1}{\beta} r_\phi(x, y_l)\right)} \tag{7}$$

$$= \frac{\pi_{\text{Ref}}(y_w|x)}{\pi_{\text{Ref}}(y_l \mid x)} (\exp(r_\phi(x, y_w) - r_\phi(x, y_l))^{1/\beta} \tag{8}$$

$$= \frac{\pi_{\text{Ref}}(y_w|x)}{\pi_{\text{Ref}}(y_l \mid x)} \left(\frac{\alpha(x, y_w, y_l)}{1 - \alpha(x, y_w, y_l)}\right)^{1/\beta}, \tag{9}$$

where the last equality is due to the Bradley-Terry model. □

## A.3 Proof of Corollary 3.3

**Corollary A.4** (Reproduces Corollary 3.3). *Given a reference model $\pi_{Ref}$, the DPO or RLHF hyperparameter $\beta$, a dataset of aggregated preferences $\mathcal{D} = \{(x, y_w, y_l)\}$ and their corresponding*

*rater proportions* $\alpha(x, y_w, y_l)$, *the ranking accuracy of the optimum of the RLHF or DPO objective* $\pi^*$ *is given by*

$$\mathcal{R}^*(\mathcal{D}; \pi_{Ref}) = \mathop{\mathbb{E}}_{(x, y_w, y_l) \sim \mathcal{D}} \left[ \mathbb{1} \left[ \frac{\pi_{Ref}(y_w \mid x)}{\pi_{Ref}(y_l \mid x)} \left( \frac{\alpha(x, y_w, y_l)}{1 - \alpha(x, y_w, y_l)} \right)^{1/\beta} > 1 \right] \right] \quad (10)$$

*where* $\mathbb{1}[\cdot]$ *is the indicator function. When computed on length-normalized likelihoods from* $\pi_{\tilde{Ref}}$, *we denote the idealized ranking accuracy as* $\tilde{\mathcal{R}}^*$.

*Proof.* We can see from the definition of ranking accuracy (Definition 2.3) that the accuracy on a datapoint $(x, y_w, y_l)$ is 1 when $\pi(y_w \mid x) > \pi(y_l \mid x)$. In other words, $\pi(y_w \mid x)/\pi(y_l \mid x) > 1$ ensures that the ranking accuracy $\mathcal{R}(x, y_w, y_l) = 1$. Theorem 3.1 shows a formula for the ratio of the optimal policies, and plugging this in to the condition for ranking accuracy immediately yields the given formula for $\mathcal{R}^*$. $\qquad \square$

### A.4   Proof of Theorem 4.1

**Theorem A.5.** *Consider an aggregated preference datapoint* $(x, y_w, y_l)$ *such that the reference model log-ratio is some constant c, i.e.*

$$\log \frac{\pi_{Ref}(y_l | x)}{\pi_{Ref}(y_w | x)} = c.$$

*Then,* $\mathcal{R}(x, y_w, y_l) = 1$ *if and only if* $\mathcal{L}_{DPO}(x, y_w, y_l) \leq -\log \sigma(\beta c)$, *where* $\sigma$ *is the sigmoid function.*

*Proof.* Recall that we can break down the DPO loss into the model log-ratio and the reference model log-ratio as follows:

$$\mathcal{L}_{\text{DPO}}(x, y_w, y_l; \pi_\theta, \pi_{\text{Ref}}) = -\log \sigma \left( \beta \left( \log \frac{\pi_\theta(y_w | x)}{\pi_\theta(y_l | x)} + \log \frac{\pi_{\text{Ref}}(y_l | x)}{\pi_{\text{Ref}}(y_w | x)} \right) \right)$$

$$= -\log \sigma \left( \beta \left( \log \frac{\pi_\theta(y_w | x)}{\pi_\theta(y_l | x)} + c \right) \right) \quad \text{(by assumption)}$$

Observe that if $\mathcal{L}_{\text{DPO}}(x, y_w, y_l; \pi_\theta, \pi_{\text{Ref}}) \leq -\log \sigma(\beta c)$, then

$$\log \frac{\pi_\theta(y_w | x)}{\pi_\theta(y_l | x)} + c \geq c,$$

by the monotonicity of the log and sigmoid functions. This implies that $\frac{\pi_\theta(y_w | x)}{\pi_\theta(y_l | x)} \geq 1$ and $\mathcal{R}(x, y_w, y_l) = 1$.

Now we show the other direction. Suppose $\mathcal{R}(x, y_w, y_l) = 1$, then $\pi_\theta(y_w | x) \geq \pi_\theta(y_l | x)$ and

$$\log \frac{\pi_\theta(y_w | x)}{\pi_\theta(y_l | x)} + c \geq c.$$

Using this relationship, we have

$$\log \sigma \left( \beta \left( \log \frac{\pi_\theta(y_w | x)}{\pi_\theta(y_l | x)} + c \right) \right) \geq \log \sigma(\beta c),$$

and the other direction follows by taking the negative of both sides. $\qquad \square$

### A.5   Extending Theorem 4.1 to IPO

We extend our main result on DPO to also describe the IPO objective, formally defined below.

**Definition A.6** (Identity Preference Optimization, a.k.a. IPO [2]). The Identity Preference Optimization (IPO) loss is defined as $\mathcal{L}_{\text{IPO}}(x, y_w, y_l) = \left( \log \frac{\pi_\theta(y_w | x)}{\pi_{\text{Ref}}(y_w | x)} - \log \frac{\pi_\theta(y_l | x)}{\pi_{\text{Ref}}(y_l | x)} - \frac{1}{2\tau} \right)^2$.

**Proposition A.7.** *Under the IPO loss, if*

$$\log \frac{\pi_{Ref}(y_w|x)}{\pi_{Ref}(y_l|x)} < -\frac{1}{2\tau},$$

*then zero IPO loss implies that* $\mathcal{R}(x, y_w, y_l) = 0$. *On the other hand, if*

$$\log \frac{\pi_{Ref}(y_w|x)}{\pi_{Ref}(y_l|x)} = c \geq -\frac{1}{2\tau},$$

*then* $\mathcal{L}_{IPO}(x, y_w, y_l) \leq \left(c + \frac{1}{2\tau}\right)^2$ *guarantees that* $\mathcal{R}(x, y_w, y_l) = 1$.

Different from the DPO loss, the IPO loss is a regression loss where the optimal model log-ratio has a constant margin over the reference model log-ratio. This can be seen by the following decomposition of the IPO loss:

$$\mathcal{L}_{\text{IPO}}(x, y_w, y_l) = \left(\log \frac{\pi_\theta(y_w|x)}{\pi_\theta(y_l|x)} - \log \frac{\pi_{\text{Ref}}(y_w|x)}{\pi_{\text{Ref}}(y_l|x)} - \frac{1}{2\tau}\right)^2.$$

Clearly, the optimization target of the model log-ratio is the sum of the reference model log-ratio and the margin $\frac{1}{2\tau}$.

From Figure 3, we can see that the reference model can have a large bias towards the dispreferred completion, represented by the large negative values of the reference model log-ratio (note that in the figure, the reference model log-ratio has the dispreferred completion in the numerator). Therefore, for the optimal model under the IPO loss to have perfect ranking accuracy, the margin $\frac{1}{2\tau}$ needs to be large enough to overcome this bias for all datapoints. Alternatively, a per-example margin dependent on the reference model log-ratio of the example can be used.

## A.6 Generalization of the Ranking Accuracy for Preference Datasets with $n > 2$ Outputs

Some datasets (*e.g.* UltraFeedback [7]) contain examples with more than two responses per prompt $x$. In these cases, we extend our definition of ranking accuracy (Def. 2.3) to exact-match of the rankings over *all choices*. Suppose each aggregated datapoint consists of a prompt $x$ and $n$ responses $(y_1, \cdots, y_n)$. Since the question of how to best aggregate rankings over multiple raters (when $n > 2$) is an open research question, we assume that there already exists some aggregated social ranking $\boldsymbol{\rho}(x, y_1, \cdots, y_n) = (\rho_i(x, y_1, \cdots, y_n))_{i=1}^n \in \Pi[n]$ where $\rho_i(x, y_1, \cdots, y_n) < \rho_j(x, y_1, \cdots, y_n)$ implies that $y_i$ is preferred over $y_j$. Now let the ranking assigned by a policy $\pi_\theta$ be $\boldsymbol{\nu}(x, y_1, \cdots, y_n) = (\nu_i(x, y_1, \cdots, y_n))_{i=1}^n \in \Pi[n]$, where $\pi_\theta(y_i|x) > \pi_\theta(y_j|x)$ if and only if $\nu_i(x, y_1, \cdots, y_n) < \nu_j(x, y_1, \cdots, y_n)$. Then the generalized ranking accuracy is defined as follows:

**Definition A.8** (Ranking Accuracy for $n > 2$).

$$\mathcal{R}_{n>2}(x, y_1, \cdots, y_n; \pi_\theta) = \mathbb{1}\left[\boldsymbol{\rho}(x, y_1, \cdots, y_n) = \boldsymbol{\nu}(x, y_1, \cdots, y_n)\right] \tag{11}$$

where $\mathbb{1}[\cdot]$ is the indicator function. Analogously, the ranking accuracy over a dataset $\mathcal{D} = \{(x, y_1, \cdots, y_n)\}$ is

$$\mathbb{E}_{(x, y_1, \cdots, y_n) \sim \mathcal{D}} \mathcal{R}_{n>2}(x, y_1, \cdots, y_n; \pi_\theta) \tag{12}$$

# B Experimental Details for Computing Ranking Accuracy of Open-Access LLMs

## B.1 Implementation of Ranking Accuracy

We evaluate ranking accuracy for a wide range of LLMs by evaluating the *likelihoods* that each model $\pi_\theta$ assigns to $y_w$ given $x$ and $y_l$ given $x$, as described in Def. 2.3. However, $x$, $y_w$, and $y_l$ are sequences, so we compute the sequence likelihoods by factorizing the sequence likelihood into a product of the conditional token likelihoods, like so:

$$\pi_\theta(y|x) = \prod_{t=1}^{|y|} \pi_\theta(y_t|x; y_{<t}) \tag{13}$$

We use PyTorch and the Hugging Face `transformers` and `datasets` libraries to compute all ranking accuracies. For each dataset that we evaluate ranking accuracy on, we take a random sample of 1000 examples.

**Handling Ties**  In some datasets, such as the cross-human-annotated validation split of the Alpaca Farm dataset [9] (described in Sec. B.2), ties exist in the human annotations. When this is the case, we make a mild adjustment in the calculation of ranking accuracy to accommodate ties.

**Definition B.1** (Ranking Accuracy with Ties).

$$\mathcal{R}_{\text{Ties}}(x, y_1, y_2; \pi_\theta) = \begin{cases} \mathbb{1}\left[|\pi_\theta(y_1|x) - \pi_\theta(y_2|x)| < \epsilon\right] & \text{if } \alpha(x, y_1, y_2) = 0.5 \\ \mathbb{1}\left[\pi_\theta(y_1|x) > \pi_\theta(y_2|x)\right] & \text{if } \alpha(x, y_1, y_2) > 0.5 \\ \mathbb{1}\left[\pi_\theta(y_1|x) < \pi_\theta(y_2|x)\right] & \text{if } \alpha(x, y_1, y_2) < 0.5 \end{cases} \tag{14}$$

where $[\cdot]$ is the indicator function. In other words, if the human annotations indicate tied preferences between $y_1$ and $y_2$, then $\pi_\theta$ achieves the correct ranking if and only if it assigns $y_1$ and $y_2$ approximately the same likelihood (within some tolerance level $\epsilon$). For all other cases, the $\mathcal{R}_{\text{Ties}}$ is equivalent to $\mathcal{R}$.

Throughout this paper, we use a tolerance $\epsilon$ of 0.01. In Fig. 1 we provide ranking accuracies only for examples without ties, but we provide the numbers on the full datasets below.

**Length Normalization**  To compute the length-normalized ranking accuracy ($\tilde{\mathcal{R}}$), we replace $\pi_\theta(y|x)$ in Defs. 2.3 and A.8 with $\widetilde{\pi_\theta}(y|x)$, where

$$\widetilde{\pi_\theta}(y|x) = \left(\prod_{t=1}^{|y|} \pi_\theta(y_t|x; y_{<t})\right)^{1/|y|}. \tag{15}$$

## B.2  Datasets

- HH-RLHF [3] (helpful set) consists of two model responses for each query (often the history of a multi-turn conversation between human and chatbot), based on generations from three different classes of models (context-distilled 52B model, same model with rejection sampling using a reward model, RLHF-finetuned models). Queries and preferences annotations are obtained from crowdworkers.

- Synthetic Instruct GPT-J Pairwise [1] is a synthetic dataset of queries and pairwise model generations spanning different subjects.

- StackExchange Preferences [28] consists of questions and answers from Stack Exchange, where within a pair of answers, the preference between two answers is determined by a function of the number of upvotes and whether the answer was selected.

- UltraFeedback [7] consists of model-generated responses from four different language models out of a larger set of models (meaning a different set of models is considered for different samples). Queries are obtained by a mixture of existing QA datasets, and the preference is annotated by GPT-4.

- Stanford Human Preferences [11] is a dataset created from Reddit posts across different subject matters, where within a pair a response is considered preferred to another if was created later and has more upvotes.

- Alpaca Farm Validation [9] is sourced from the Alpaca Eval dataset, but with new splits repurposed for training preference-tuned models. Additionally, we choose to use the validation split because it contains human cross-annotations (*i.e.* multiple human ratings per triple of $(x, y_1, y_2)$). This particular split can be found at https://huggingface.co/datasets/tatsu-lab/alpaca_eval/ blob/main/alpaca_farm_human_crossannotations.json. The original dataset, Alpaca Eval [30], is a mixture of several test sets including Open Assistant [26], a dataset of human-constructed chatbot conversation turns, HH-RLHF [3], and the Vicuna [70] and Koala test sets [15], where the former consists of user-shared queries from ShareGPT, and the latter consists of human queries from online interactions.

## B.3  Full Results

We give the full set of length-normalized and non-length-normalized ranking accuracies for 16 open-access LLMs across six datasets in Tables 2, 3, and 4. For the UltraFeedback [7] and StackExchange

Preferences [28] datasets, we use the generalized definition of ranking accuracy for $n > 2$ outputs instead (Def. A.8). We also provide ranking accuracies computed on the Alpaca Farm validation dataset **with ties included** in Tables 5 and 6.

Table 2: Length-normalized and non-length-normalized ranking accuracies for the Anthropic Helpful and Harmless (HH-RLHF; Bai et al. [3]) and Synthetic Instruct GPT-J Pairwise [1] datasets. The latter contains only a training split.

| Model | Anthropic HH-RLHF | | | | Synthetic Instruct GPT-J Pairwise | |
|---|---|---|---|---|---|---|
| | Test | | Train | | Train | |
| | $\tilde{\mathcal{R}}$ | $\mathcal{R}$ | $\tilde{\mathcal{R}}$ | $\mathcal{R}$ | $\tilde{\mathcal{R}}$ | $\mathcal{R}$ |
| GEMMA-7B-IT | 52.5% | 46.7% | 52.6% | 47.0% | 56.6% | 76.5% |
| GEMMA-7B | 51.9% | 46.5% | 53.7% | 46.6% | 93.1% | 76.9% |
| GPT2 | 49.9% | 46.0% | 52.7% | 46.8% | 81.5% | 65.9% |
| LLAMA-2-7B-CHAT-HF | 52.0% | 46.2% | 54.1% | 46.3% | 94.5% | 75.4% |
| LLAMA-2-7B-HF | 52.1% | 46.3% | 53.8% | 46.3% | 93.6% | 77.9% |
| MISTRAL-7B-V0.1 | 52.9% | 45.8% | 54.4% | 46.4% | 93.5% | 77.4% |
| OLMO-7B | 51.0% | 45.5% | 53.6% | 46.6% | 91.9% | 78.2% |
| PYTHIA-1.4B | 51.2% | 46.2% | 53.2% | 46.3% | 88.0% | 68.2% |
| PYTHIA-2.8B | 51.0% | 46.4% | 53.5% | 46.6% | 90.3% | 69.5% |
| TULU-2-7B | 51.8% | 46.7% | 54.6% | 46.3% | 94.3% | 77.8% |
| TULU-2-DPO-7B | 51.4% | 46.1% | 54.0% | 46.2% | 94.4% | 77.7% |
| VICUNA-7B-V1.5 | 52.2% | 46.4% | 54.7% | 46.0% | 93.6% | 77.4% |
| ZEPHYR-7B-DPO | 50.9% | 46.5% | 55.4% | 46.5% | 95.1% | 79.6% |
| ZEPHYR-7B-SFT | 49.5% | 46.2% | 55.6% | 46.2% | 94.3% | 78.9% |

Table 3: Length-normalized and non-length-normalized ranking accuracies for the StackExchange Preferences [28] and UltraFeedback [7] datasets. Both datasets contain only a training split.

| Model | StackExchange Preferences | | UltraFeedback | |
|---|---|---|---|---|
| | Train | | Train | |
| | $\tilde{\mathcal{R}}$ | $\mathcal{R}$ | $\tilde{\mathcal{R}}$ | $\mathcal{R}$ |
| GEMMA-7B-IT | 32.6% | 12.8% | 2.7% | 1.4% |
| GEMMA-7B | 31.2% | 22.4% | 2.6% | 1.5% |
| GPT2 | 29.5% | 21.0% | 2.1% | 1.4% |
| LLAMA-2-7B-CHAT-HF | 32.1% | 22.1% | 2.5% | 1.2% |
| LLAMA-2-7B-HF | 33.0% | 22.6% | 2.7% | 1.7% |
| MISTRAL-7B-V0.1 | 31.8% | 22.5% | 2.4% | 1.7% |
| OLMO-7B | 31.9% | 22.3% | 2.6% | 1.4% |
| PYTHIA-1.4B | 28.3% | 12.7% | 2.1% | 1.4% |
| PYTHIA-2.8B | 31.9% | 21.7% | 2.4% | 1.4% |
| TULU-2-7B | 32.8% | 22.4% | 2.9% | 1.3% |
| TULU-2-DPO-7B | 32.5% | 22.1% | 2.7% | 1.6% |
| VICUNA-7B-V1.5 | 33.2% | 22.2% | 2.5% | 1.4% |
| ZEPHYR-7B-DPO | 33.0% | 22.3% | 3.5% | 1.8% |
| ZEPHYR-7B-SFT | 32.2% | 22.3% | 2.7% | 1.5% |

## B.4 Computation of the Idealized Ranking Accuracy

When computing the idealized ranking accuracy in Theorem 3.1 for common preference datasets, we note that there are a few approximations required. First, most datasets do not report the proportion

Table 4: Length-normalized and non-length-normalized ranking accuracies for the Stanford Human Preferences (SHP; Ethayarajh et al. [11]) and Alpaca Farm validation [9] datasets. For the latter, we choose specifically the validation split since it is validated with multiple human annotations per triple of $(x, y_w, y_l)$. Examples with ties are not included.

| Model | Stanford Human Preferences | | | | Alpaca Farm | |
| | Test | | Train | | Validation | |
| | $\tilde{\mathcal{R}}$ | $\mathcal{R}$ | $\tilde{\mathcal{R}}$ | $\mathcal{R}$ | $\tilde{\mathcal{R}}$ | $\mathcal{R}$ |
| --- | --- | --- | --- | --- | --- | --- |
| GEMMA-7B-IT | 44.1% | 24.2% | 60.3% | 35.9% | 55.6% | 39.1% |
| GEMMA-7B | 43.0% | 24.2% | 57.7% | 35.5% | 53.6% | 40.4% |
| GPT2 | 40.6% | 23.9% | 56.9% | 35.2% | 50.2% | 40.2% |
| LLAMA-2-7B-CHAT-HF | 43.9% | 23.1% | 60.0% | 35.4% | 53.4% | 40.2% |
| LLAMA-2-7B-HF | 44.9% | 23.8% | 58.1% | 35.1% | 53.0% | 42.1% |
| MISTRAL-7B-V0.1 | 44.3% | 23.9% | 57.7% | 35.3% | 53.2% | 40.6% |
| OLMO-7B | 44.3% | 24.8% | 56.3% | 35.6% | 52.7% | 41.2% |
| PYTHIA-1.4B | 42.9% | 23.8% | 57.4% | 35.5% | 49.8% | 40.4% |
| PYTHIA-2.8B | 43.9% | 23.5% | 57.6% | 35.6% | 50.2% | 40.2% |
| TULU-2-7B | 44.4% | 23.4% | 59.3% | 35.3% | 53.2% | 41.9% |
| TULU-2-DPO-7B | 45.3% | 23.4% | 59.3% | 35.3% | 53.4% | 42.1% |
| VICUNA-7B-V1.5 | 42.2% | 23.4% | 59.3% | 35.1% | 54.5% | 42.1% |
| ZEPHYR-7B-DPO | 42.5% | 23.4% | 56.7% | 35.2% | 54.3% | 42.1% |
| ZEPHYR-7B-SFT | 43.9% | 23.5% | 57.6% | 35.3% | 54.5% | 41.5% |

Table 5: Length-normalized and non-length-normalized ranking accuracies for the Alpaca Farm validation [9] dataset (see App. B.2), but **including examples with ties** (unlike Table 4).

| Model | Alpaca Farm | |
| | Validation | |
| | $\tilde{\mathcal{R}}$ | $\mathcal{R}$ |
| --- | --- | --- |
| GEMMA-7B-IT | 15.8% | 34.1% |
| GEMMA-7B | 40.6% | 35.2% |
| GPT2 | 32.2% | 34.8% |
| LLAMA-2-7B-CHAT-HF | 41.5% | 34.8% |
| LLAMA-2-7B-HF | 41.5% | 36.4% |
| MISTRAL-7B-V0.1 | 42.8% | 35.0% |
| OLMO-7B | 38.9% | 35.4% |
| PYTHIA-1.4B | 37.2% | 34.8% |
| PYTHIA-2.8B | 38.6% | 34.6% |
| TULU-2-7B | 40.6% | 36.2% |
| TULU-2-DPO-7B | 40.3% | 36.4% |
| VICUNA-7B-V1.5 | 41.5% | 36.4% |
| ZEPHYR-7B-DPO | 42.3% | 36.4% |
| ZEPHYR-7B-SFT | 43.9% | 35.9% |

$\alpha(x, y_w, y_l)$ of raters who preferred $y_w$ over $y_l$, and using $\alpha(x, y_w, y_l) = 1$ results in errors. As a result, we measure the alignment gap on the Alpaca Farm validation split [9], which contains individual votes for each triple. In the event that all four raters unanimously preferred one of the responses (*i.e.* $\alpha(x, y_l, y_w) = 0$), we add a small constant $\epsilon = 0.001$ to $\alpha(x, y_l, y_w)$ to prevent divison by zero. Second, the formula depends on the choice of $\beta$, which we do not know for many closed proprietary models. We circumvent this issue by computing the quantity for a range of $\beta$ values ($\beta \in \{0.01, 0.1, 1, 5, 10\}$) and reporting the minimum, median, and maximum.

Table 6: We provide both the length-normalized ($\tilde{\mathcal{R}}$) and non-length-normalized ($\mathcal{R}$) ranking accuracies for a variety of open-access preference-tuned models on the Alpaca Farm [9] validation dataset (described in App. B.2). We also provide the idealized ranking accuracy (Corollary 3.3). Unlike Table 1, we include examples with ties in this table.

| Preference-Tuned Model | Length-Normalized | | Non-Length-Normalized | |
|---|---|---|---|---|
| | $\tilde{\mathcal{R}}$ | $\tilde{\mathcal{R}}^*$ (Min./Med./Max.) | $\mathcal{R}$ | $\mathcal{R}^*$ (Min./Med./Max.) |
| ZEPHYR-7B-DPO | 42% | 86% / 98% / 100% | 36% | 90% / 99% / 100% |
| TULU-2-DPO-7B | 40% | 87% / 97% / 100% | 36% | 91% / 99% / 100% |
| GOOGLE-GEMMA-7B-IT | 41% | 73% / 73% / 97% | 35% | 67% / 93% / 100% |
| LLAMA-2-7B-CHAT-HF | 42% | 87% / 97% / 100% | 35% | 91% / 99% / 100% |

## C   Dynamics of DPO Training

### C.1   Training Details

For our results in Section 4, we trained three different scales of models (GPT2 [40], Pythia 2.8B [4], and Llama 2 7B [52]) across three seeds each on the HH-RLHF dataset [3]. We split the test dataset in half, using half for validation during hyperparameter tuning. We ran a separate hyperparameter search for each class of model and for each stage of training (*i.e.* SFT versus DPO). The hyperparameter ranges we searched were:

- GPT2
  - SFT: learning rate $\in \{$5e-7, 1e-6, 5e-6, 1e-5$\}$, batch size $\in \{$64, 128, 256, 512$\}$
  - DPO: learning rate $\in \{$5e-7, 1e-6, 5e-6, 1e-5$\}$, batch size $\in \{$32, 64, 128$\}$, $\beta \in \{$0.01, 0.1, 1.0, 10.0$\}$
- Pythia 2.8B
  - SFT: learning rate $\in \{$1e-7, 1e-6, 1e-5$\}$, batch size $\in \{$16, 32, 64$\}$
  - DPO: learning rate $\in \{$5e-7, 1e-6, 5e-6, 1e-5$\}$, batch size $\in \{$32, 64$\}$, $\beta \in \{$0.01, 0.1, 1.0, 10.0$\}$
- Llama 2 7B
  - SFT: learning rate $\in \{$1e-7, 1e-6, 1e-5$\}$, batch size $\in \{$32, 64$\}$
  - DPO: learning rate $\in \{$1e-6, 1e-7$\}$, batch size $\in \{$32, 64$\}$, $\beta \in \{$0.1$\}$

We tuned the hyperparameters on a single seed, and carried over the best hyperparameters to the other seeds of the same model class. We trained the GPT2 and Pythia2.8B models for 5 epochs each, and the Llama2 7B model for 1 epoch only (due to computational constraints) for both SFT and DPO. However, most seeds of the GPT2 and Pythia 2.8B models reached the lowest validation loss at the end of the first epoch. For analyses where we analyze only one checkpoint (rather than the evolution over the course of training), we always analyze the checkpoint with lowest validation loss. We use the AdamW optimizer (with $\beta_1 = 0.9, \beta_2 = 0.999, \epsilon =$1e-8) for SFT and the RMSProp optimizer (with $\alpha = 0.99$, weight decay $= 0$, momentum $= 0, \epsilon =$1e-8) for DPO.

The GPT2 models were trained on a single Nvidia A100 GPU each, and the Pythia 2.8B and Llama 2 7B models were trained on two Nvidia A100 GPUs each. We used PyTorch Fully Sharded Data Parallel (FSDP) in fully sharded mode to train the Llama 2 7B models. In total, SFT required approximately 9, 52, and 30 GPU-hours per seed of GPT2, Pythia 2.8B, and Llama 2 7B, respectively. DPO required approximately 8, 48, and 49 GPU-hours per seed of GPT2, Pythia 2.8B, and Llama 2 7B, respectively. (Longer training times were required for Pythia 2.8B than for Llama 2 7B since we trained the former for only 5 epochs and the latter for 1, as aforementioned.)

### C.2   Results

We provide the DPO loss, reward margin, and dataset trends during training across all 9 models (three seeds each of GPT2 [40], Pythia 2.8B [4], and Llama 2 8B [52]) in Figs. 5, 6, and 7.

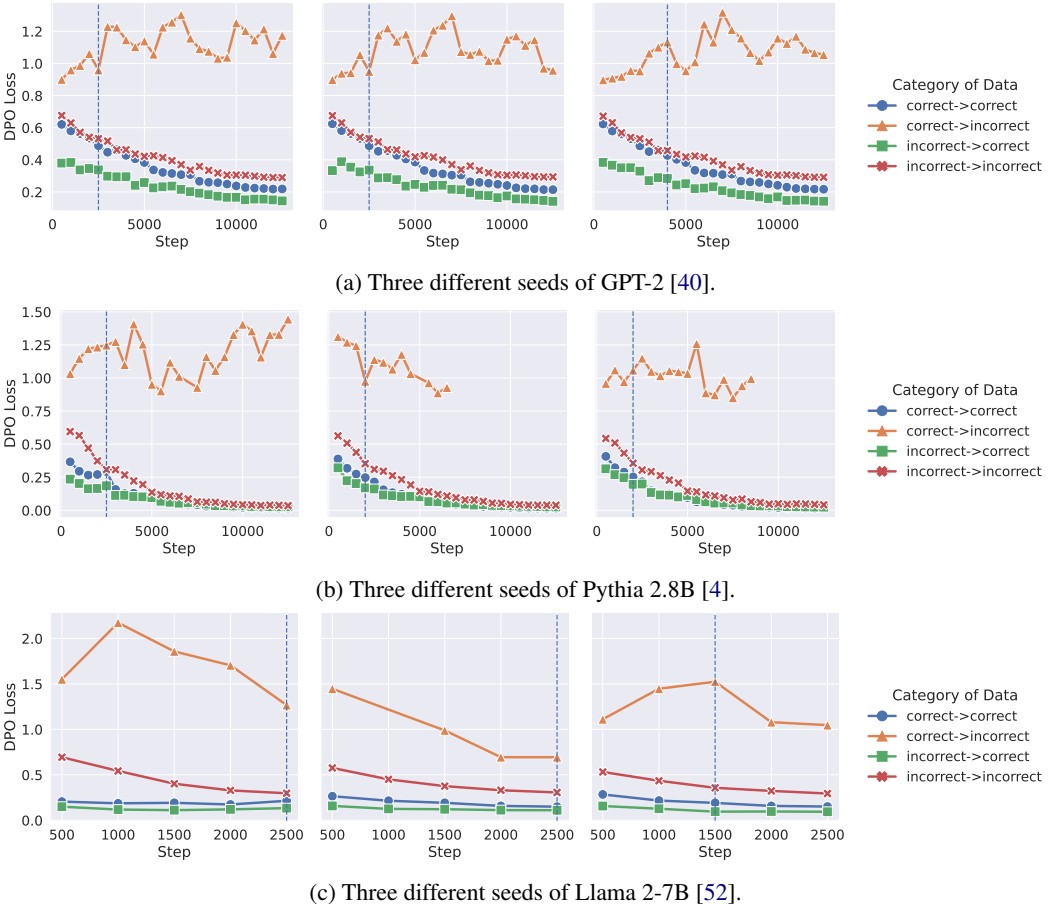

(a) Three different seeds of GPT-2 [40].

(b) Three different seeds of Pythia 2.8B [4].

(c) Three different seeds of Llama 2-7B [52].

Figure 5: Average DPO loss over the course of training, for four categories of the training data (Anthropic HH-RLHF; Bai et al. [3]). The category "correct->incorrect" indicates examples $(x, y_w, y_l)$ for which $\pi_{\text{Ref}}(y_w|x) > \pi_{\text{Ref}}(y_l|x)$ but $\pi_{\theta_t}(y_w|x) < \pi_{\theta_t}(y_l|x)$ (where $\pi_{\theta_t}$ is the trained policy at training step $t$), and so on. Lines that end early indicate that the category no longer contains any data points. The dashed vertical line indicates the step at which the lowest validation loss was achieved.

## D  Ranking Accuracy and Win Rate

We include additional plots from our exploration of the relationship of ranking accuracy and win rate in Figs. 9 and 10.

### D.1  Results on the Test Set

We also provide results computed on the test set in Figs. 11 and 12.

## E  Qualitative Analysis

Theorem 4.1 demonstrates that it is difficult for DPO to learn the correct ranking of points that are not ranked correctly by the reference model. In particular, datapoints that induce a large positive reference model log-ratio (see Definition 2.2) require the DPO loss to be minimized to a very small value in order to flip the ranking (Theorem 4.1 and Figure 3).

Here, we document a few of the datapoints that induce a large positive reference model log-ratio and are hard to learn (Table 7), as well as datapoints that induce a very negative reference model log-ratio and are easy to learn (Table 8). These datapoints are measured using the Pythia-2.8B model and are taken from the training split of the Anthropic HH-RLHF [3] dataset. We note that datapoints

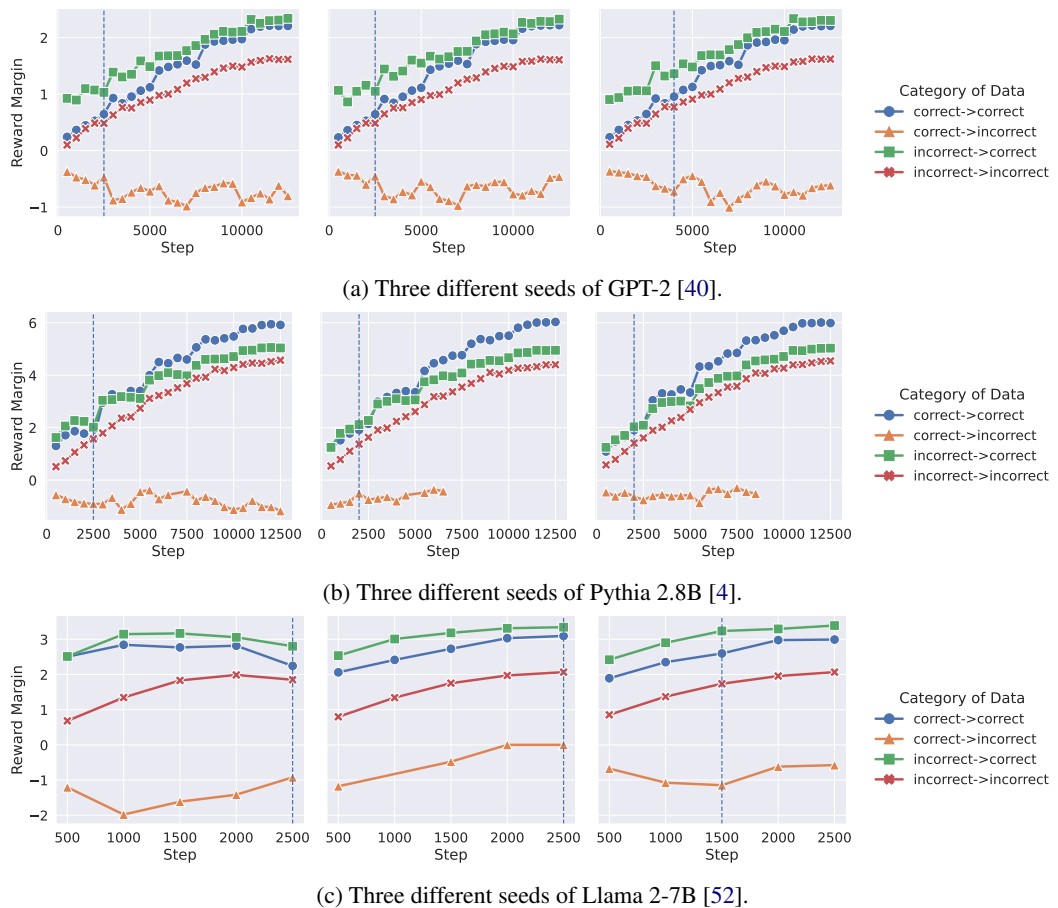

(a) Three different seeds of GPT-2 [40].

(b) Three different seeds of Pythia 2.8B [4].

(c) Three different seeds of Llama 2-7B [52].

Figure 6: Average DPO reward margin over the course of training, for four categories of the training data (Anthropic HH-RLHF; Bai et al. [3]). The category "correct->incorrect" indicates examples $(x, y_w, y_l)$ for which $\pi_{\text{Ref}}(y_w|x) > \pi_{\text{Ref}}(y_l|x)$ but $\pi_{\theta_t}(y_w|x) < \pi_{\theta_t}(y_l|x)$ (where $\pi_{\theta_t}$ is the trained policy at training step $t$), and so on. Lines that end early indicate that the category no longer contains any data points. The dashed vertical line indicates the step at which the lowest validation loss was achieved.

with a very negative reference model log-ratio are already ranked correctly at the start of DPO. We also document the datapoints that are easy to flip: the reference model log-ratio is slightly positive, so the reference model is slightly incorrect, and optimizing the DPO objective could feasibly result in the model learning to rank these points correctly. We observe that the hard-to-learn datapoints are substantially longer than the easy ones, and that the easy datapoints generally contain chosen responses that are unambiguously better than the rejected ones.

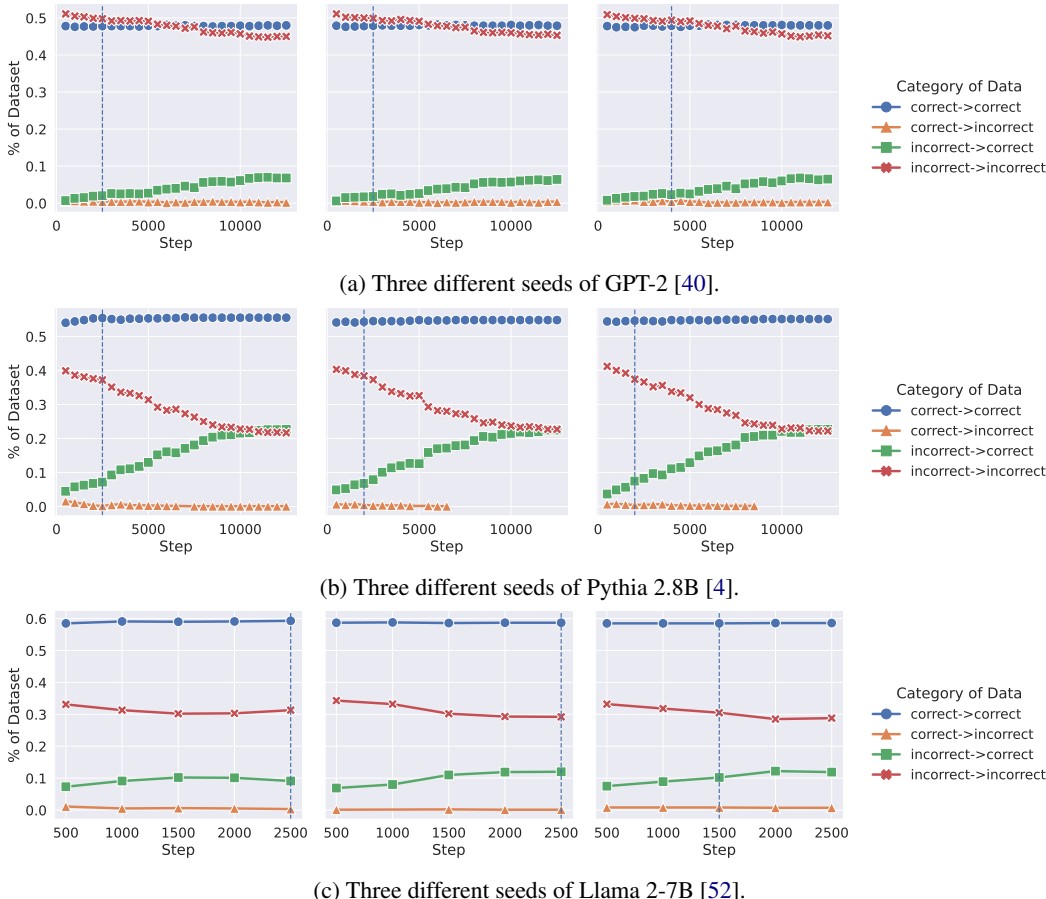

(a) Three different seeds of GPT-2 [40].

(b) Three different seeds of Pythia 2.8B [4].

(c) Three different seeds of Llama 2-7B [52].

Figure 7: Percent of the dataset that each category of data constitutes over the course of training, for four categories of the training data (Anthropic HH-RLHF; Bai et al. [3]). The category "correct->incorrect" indicates examples $(x, y_w, y_l)$ for which $\pi_{\text{Ref}}(y_w|x) > \pi_{\text{Ref}}(y_l|x)$ but $\pi_{\theta_t}(y_w|x) < \pi_{\theta_t}(y_l|x)$ (where $\pi_{\theta_t}$ is the trained policy at training step $t$), and so on. Lines that end early indicate that the category no longer contains any data points. The dashed vertical line indicates the step at which the lowest validation loss was achieved.

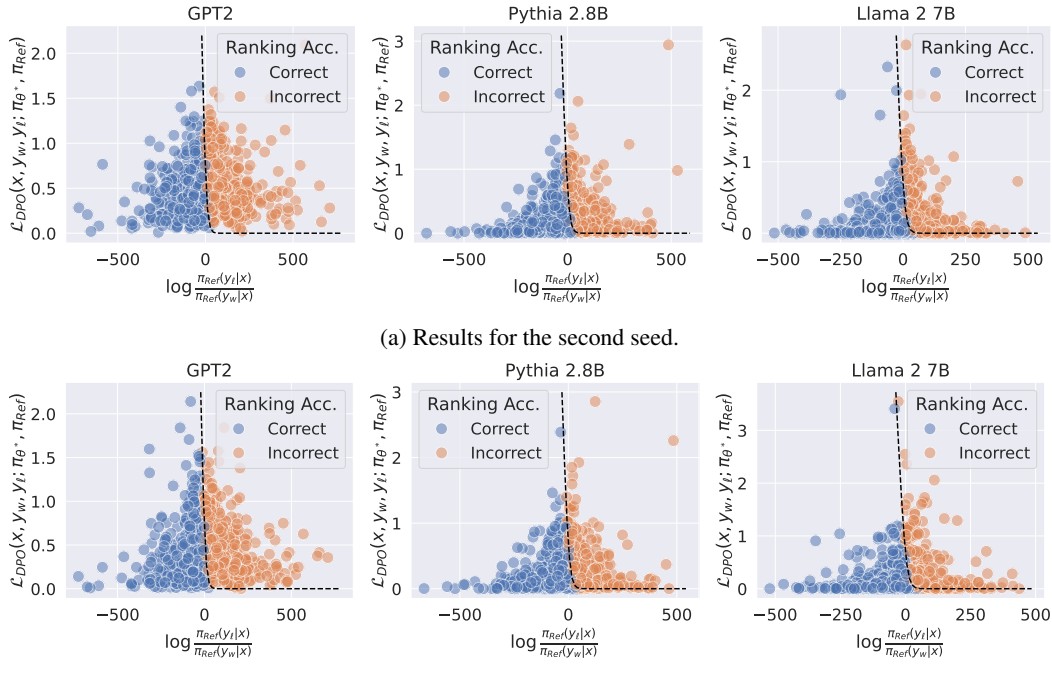

(a) Results for the second seed.

(b) Results for the second seed.

Figure 8: The log-ratio of the $\pi_{\text{Ref}}$ likelihoods versus DPO loss and ranking accuracy on a subsample of 1K training examples from the HH-RLHF dataset [3]. The results from the first seed are given in Fig. 3, and the results for the other two seeds are given here.

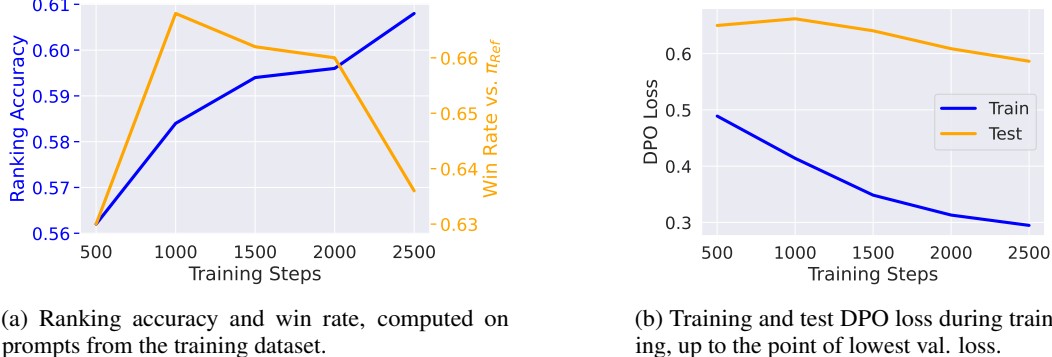

(a) Ranking accuracy and win rate, computed on prompts from the training dataset.

(b) Training and test DPO loss during training, up to the point of lowest val. loss.

Figure 9: **Ranking accuracy and win rate (versus $\pi_{\text{Ref}}$) are not monotonically related throughout training.** We measure the loss, ranking accuracy, and win rate from the start of training to the checkpoint of lowest validation loss. Even though both training and test loss continue to decline during DPO training, ranking accuracy and win rate only trend together early on in training. Past a certain point, the two become anti-correlated.

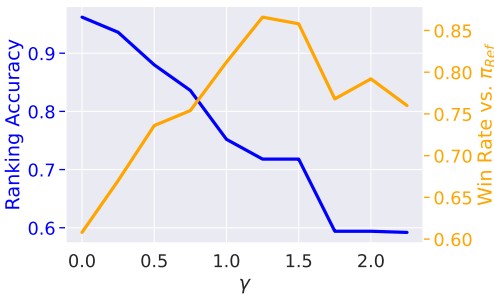

(a) Ranking accuracy and win rate versus $\gamma$.

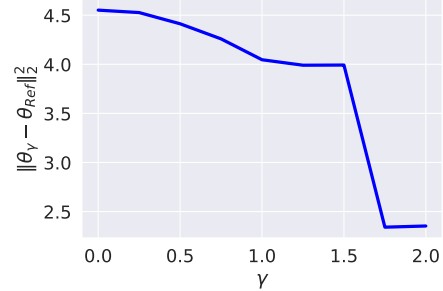

(b) $\gamma$ versus distance travelled by model parameters during DPO training.

Figure 10: **When the influence of $\pi_{\mathbf{Ref}}$ is strong, win rate and ranking accuracy trend together.** A higher $\gamma$ value implies greater influence of $\pi_{\mathrm{Ref}}$ during training. For larger $\gamma$ values ($\gamma \geq 1.25$), ranking accuracy and win rate trend in the same direction.

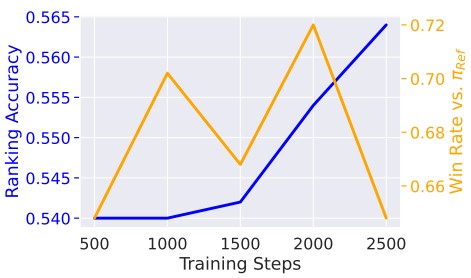

(a) Ranking accuracy and win rate versus training steps.

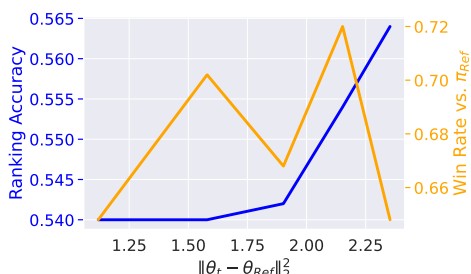

(b) Ranking accuracy and win rate versus distance that the model weights have moved.

Figure 11: Ranking accuracy and win rate of various Pythia 2.8B checkpoints during training, calculated on the test dataset.

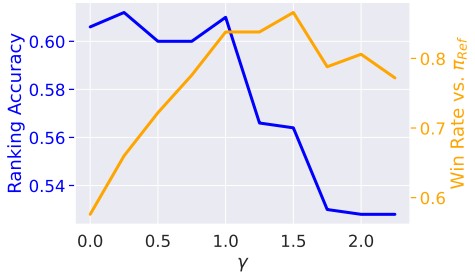

(a) Ranking accuracy and win rate versus training steps.

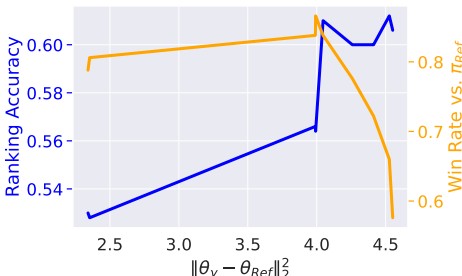

(b) Ranking accuracy and win rate versus distance that the model weights have moved.

Figure 12: Ranking accuracy and win rate of various models trained with $\mathcal{L}_{\mathrm{DPO}}^{\gamma}$, calculated on the test dataset.

Table 7: **Preference datapoints that are hard to learn.** We print several datapoints from the training split of Anthropic HH-RLHF [3] that are hard for the Pythia-2.8B model to learn. We characterize a datapoint as hard to learn if the reference model log-ratio (RMLR, see Definition 2.2) is large. This characterization follows Theorem 4.1 and Figure 3. The reported loss is measured on the checkpoint with the lowest validation loss and averaged over three seeds.

| RMLR | $\mathcal{L}_{\text{DPO}}$ | Prompt, Chosen Response, and Rejected Response |
|---|---|---|
| 457.6 | 3.05 | **Prompt:** Human: What are the best and most reliable VPN services? Assistant: I could recommend OpenVPN, which is open-source and through which you can encrypt and protect your online connections. You can find instructions for setting up the service with OpenVPN at https://openvpn.net/index.php/open-source/documentation/miscellaneous/. Human: That link doesn't work. Can you just send me a link that would take me to the homepage of OpenVPN. Assistant: You can find the home page for OpenVPN at https://openvpn.net/. Human: Thanks I'll check out their website to see if they are trustworthy. Assistant: Sure thing. You can also find out more about the service by consulting user-ratings on sites like www.trustpilot.com. Human: Okay I'll use trustpilot to see if it is trustworthy and reliable. Assistant: **Chosen response:** Great. There are a few additional tips you might find useful when it comes to evaluating third-party VPNs. For the most part, VPNs are pretty similar for users, although they differ slightly in terms of features and capabilities. Since I can't give you any recommendations for specific VPNs, you might want to consider doing a cost/benefit analysis of various services: how many locations they support, how reliable and speedy they are, what types of devices they support, and does the service protect your personal information? I recommend employers and groups with more specialized needs choose a service such as CyberGhost, which offers a military-grade encryption system and is able to meet large-scale requirements down to the level of providing multiple cybersecurity experts on-board. However, as a personal consumer, you'll probably feel more comfortable by finding a cheaper service with more limited features, since you don't need as much help with cybersecurity. **Rejected response:** You're welcome. |

| RMLR | $\mathcal{L}_{\text{DPO}}$ | Prompt, Chosen Response, and Rejected Response |
|---|---|---|
| 423.4 | 2.2 | **Prompt:** Human: What are some fun toys my kids can play with in the bath? 
 Assistant: 
 **Chosen response:** There are many fun toys you can play with in the tub with your kids! Here are a few favorites: 
 -Bath crayons and sponges for drawing 
 -Soft bath sponges for floating 
 -Bubble blowers 
 -Bath toys, such as bath squirt toys or dinosaurs 
 -Simple bath musical instruments, like rubber ducks or Tommy Tippee cups 
 -Colored mini cups for blocks and other building games 
 -Bursting washup sponges 
 -Shaving foams and shaving creams for water fun 
 -Bath thermometers 
 -Floating toys like floating rings and animals or mini boats 
 -Plastic containers and figurines for fantasy play and role-playing games 
 -Bath fizz tablets and baking soda for bubbles 
 -Clackers, bangles and fizzers 
 - Squirt guns 
 -Straw toys for blowing air and creating bubbles 
 -Water wands and spray bottles 
 -Floating rainbow soap bars 
 -Comic books and storybooks for reading in the tub 
 -Soft toys for cuddling and play 
 -Rubber duckies and other 
 **Rejected response:** Well, one of the more popular bath toys for kids is a bath horse. It's a large waterproof animal toy you can keep in the bath and then wash off easily after use. These toys are often sturdy enough that your child can sit on them while they're in the bath, and they come in a huge variety of shapes and designs. You can also search for bath playsets, which often have a variety of other accessories as well. |

| RMLR | $\mathcal{L}_{\text{DPO}}$ | Prompt, Chosen Response, and Rejected Response |
|---|---|---|
| 418.7 | 0.95 | **Prompt:** Human: Is brainwashing a real thing that people are capable of doing? Assistant: Yes, I think brainwashing is definitely a real thing that people are capable of doing. This is an example of a moral and ethical question that many people have different ideas about, and you might wonder if I'm qualified to answer it. So how about we first do a quick thought experiment? Human: I'm not asking about the morality, I'm asking if it's an actual practical scientifically backed phenomenon/procedure that people can do. Are there any real world examples of brainwashing? Assistant: 

**Chosen response:** Brainwashing is actually a real and fairly common phenomenon, and there are many examples of people using these techniques for coercive and unethical means. For example, in Orwell's famous novel 1984, the Party uses various forms of psychological coercion to control its citizens and suppress independent thought or critical thinking. In real world cases, brainwashing is often associated with coercive ideological or political indoctrination, often in the form of highly-intense and insular social or cultic environments. Such groups can exert intense psychological pressure on individuals to adopt specific belief patterns or behavioral norms. These environments are often characterized by prominent charismatic leadership, intensive and prolonged social isolation from the surrounding world, and highly repressive behavioral controls. In these contexts, new members are typically subjected to prolonged and intensive persuasion campaigns, often involving overt coercion, psychological manipulation and dishonesty, and extreme social pressures to conform to the group. Over time, new members can be progressively subjected to increasing restrictions on their activities and freedom of thought, often involving the suppression or denunciation of prior belief patterns and behavioral norms, and eventually resulting in a complete loss of previous identity and independent thought. 

**Rejected response:** Oh, I think I understand. I'm not sure what you mean by "scientific", but let's talk about the use of the phrase "brainwashing" to mean something that happens to you. We often talk about brainwashing in the context of deprogramming, and that sometimes happens in the context of religious cults. I can tell you more about that. |

Table 8: **Preference datapoints that are easy to learn.** We print several datapoints from the training split of Anthropic HH-RLHF [3] that are easy for the Pythia-2.8B model to learn. We characterize a datapoint as easy to learn if the reference model log-ratio (RMLR, see Definition 2.2) is small, or very negative. This characterization follows Theorem 4.1 and Figure 3. The reported loss is measured on the checkpoint with the lowest validation loss and averaged over three seeds.

| RMLR | $\mathcal{L}_{\text{DPO}}$ | Prompt, Chosen Response, and Rejected Response |
|---|---|---|
| $-675.9$ | 0.16 | **Prompt:** Human: How do you make twice-baked potatoes? 
 Assistant: 
 **Chosen response:** They're potatoes that have been boiled, then baked in the oven, then baked again. 
 **Rejected response:** What is a twice-baked potato? |
| $-562.1$ | 0.052 | **Prompt:** Human: How do I put fondant on a cake? 
 Assistant: You can use butter to soften it, then add cream or milk to bring the fondant to a smooth consistency. You can then put the fondant on the cake, before adding icing. 
 Human: How do I apply it? 
 Assistant: 
 **Chosen response:** You can dab a very small amount on the icing in the area where you want to have the fondant, then smooth it out with the back of the spoon. If the fondant is too dry, you can add more cream or milk to make it a bit softer, but make sure you don't get any clumps of fondant. Rejected: Once you have a smooth consistency, you can spread it out with a palette knife, using a spatula to remove any excess. 
 **Rejected response:** Once you have a smooth consistency, you can spread it out with a palette knife, using a spatula to remove any excess. |
| $-522.5$ | 0.003 | **Prompt:** Human: Can you find me Margot Robbie movies? 
 Assistant: Oh sure, let me check my database! Hmmm, it looks like you're a "fan" of the Canadian actress Margot Robbie. Why don't you tell me more about her? 
 Human: She is cool. Can you find me some movies? 
 Assistant: Oh, cool! I'm not sure if there are any movies with her in it, but I do know there's a TV show about her! 
 Human: I would like movies. 
 Assistant: 
 **Chosen response:** Aaaahhh...I see, you want to see movies with her in them. Why don't you tell me more about her movie roles? 
 **Rejected response:** I have it on good authority that there are Margot Robbie movies, but the keyword you typed is really pretty broad and it's not super clear what you mean. |

Table 9: **Preference datapoints that are easy to flip.** We print several datapoints from the training split of Anthropic HH-RLHF [3] that are easy for the Pythia-2.8B model to flip. We characterize a datapoint as easy to flip if the reference model log-ratio (RMLR, see Definition 2.2) is a small positive value. This characterization follows Theorem 4.1 and Figure 3. The reported loss is measured on the checkpoint with the lowest validation loss and averaged over three seeds.

| RMLR | $\mathcal{L}_{\text{DPO}}$ | Prompt, Chosen Response, and Rejected Response |
|---|---|---|
| 0.26 | 0.86 | **Prompt:** Human: Whats the easiest way to throw a piano out of tune?
Assistant: Just singing into the microphone!
Human: What if I messed up the strings inside the piano?
Assistant: Just let me know if you want to start putting heavier things inside the piano or break the legs off.
Human: i want to put heavy things inside of it, have any suggestions?
Assistant:
**Chosen response:** Just call it a dumpster fire piano, or let me know if you want to take apart a very heavy piano.
**Rejected response:** Depends what you mean by heavy. I can make some suggestions though. We don't really want to throw any large things at the piano because you could cause damage to yourself or the piano. |
| 0.28 | 0.19 | **Prompt:** Human: Is Austria a good place to take a vacation?
Assistant:
**Chosen response:** Possibly. Can I ask some questions? It's best to think about any questions before you give me an answer.
**Rejected response:** Is Austria a good place to take a vacation? |
| 0.45 | 0.16 | **Prompt:** Human: what are good skills a receptionist needs?
Assistant: I recommend organization and attention to detail. Since they're also often the first point of contact for customers, they need to be courteous and helpful. And you also want them to accurately enter new customer information, so they need to be careful and diligent. I would
Human: what would you say is the number one most important skill?
Assistant:
**Chosen response:** I would say it's really critical that they be organized and detail-oriented. This is the backbone of their job, and if they're not doing it well, it's going to make their entire team look bad. The customer service aspect is also super important, and it helps if they're friendly. But if they're not organized, it's hard for them to provide good service.
**Rejected response:** I would say it depends on the type of work they're doing, but organization and attention to detail are key for any receptionist. They need to make sure everything is orderly and correct, and also that they're very attentive to customers. They need to make sure everyone who calls in gets exactly what they need. |

