# OpenReview forum: "Preference Learning Algorithms Do Not Learn Preference Rankings"
_NeurIPS.cc/2024/Conference — NeurIPS 2024 poster_

### Official Review · Reviewer_uW8U · 2024-07-08

**Soundness:** 3
**Presentation:** 4
**Contribution:** 3
**Rating:** 5
**Confidence:** 4

**Summary:**

This paper investigates the effectiveness of preference learning algorithms, include the RLHF (PPO-based) and DPO in training LLMs. In particular, the authors study the non-regularized ranking accuracy, which is largely ignored in the previous RLHF literature. The authors approach this investigation through both theoretical analysis and empirical experiments and show that whether the ranking is correct or not  highly depends on the initial reference policy.

**Strengths:**

The research problem is interesting. While the RLHF and DPO mainly care about the KL-regularized target, it is debatable that the likelihood of the response under the resulting policy is what we are really interested in. Therefore, the notion of ranking policy in terms of this non-regularized target deserves more attention.

With the goal in mind, the DPO loss target in definition 2.2 clearly shows that the log-ratio on the reference model serves as an adaptive margin for the data point. Therefore, the discussions around the reference model is natural and informative.

I particularly appreciate the section 4 on the empirical studies of DPO, which can largely help us to understand the dynamic of this widely adopted algorithm.

Overall, the authors identify an interesting question and the present a complete story with solid evidence in both theory and experiments.

**Weaknesses:**

My major concern is that this paper only studies the goal of helpfulness in chat style.

First, while it is fair to say that DPO is not good at optimizing the ranking policy in terms of the non-regularized target, it is not immediately clear that the non-regularized ranking policy is a better metric. For instance, a concurrent work (so this should not hurt the novelty of this paper) [1] proposes simpo with many shared insights. In particular, they also notice the mismatch between the ranking policy and the target of DPO. However, in their experiments, as the win rates evaluated on the standard benchmark significantly increase, we also observe a significant drop in some academic benchmarks, particularly those on the mathematical problems solving.

Based on my personal experience, if we adopt a smaller KL coefficient (e.g., 0.01, so less regularization), the final model performance is worse than the model with a moderate one (say 0.1). This is particularly true for the online iterative learning, which we are more interested in as the authors also notice that these variants outperform the offline one with a large margin. In most of the cases, the ability of maximizing the reward/ranking accuracy, while maintaining a moderate KL divergence is more suitable to evaluate the algorithm performance in my viewpoint. Therefore, the ranking accuracy alone is also not a good learning target, which is also evidenced by figure 4 of this work.

Based on the above two points, I feel that the paper can still be largely improved if a more comprehensive evaluations can be done, like the trade-off between the performance on instruction-following benchmarks (alpaca-eval, mt-bench, gpt4 win rate) and other  academic benchmarks. But overall, I support the acceptance.

[1] SimPO: Simple Preference Optimization with a Reference-Free Reward

**Questions:**

see weakness

**Limitations:**

yes

---

> ### Author Rebuttal · Authors · 2024-08-06
>
> We thank the reviewer for their detailed review and greatly appreciate their support of the acceptance of our paper. We address the feedback below:
>
> **"First, while it is fair to say that DPO is not good at optimizing the ranking policy in terms of the non-regularized target, it is not immediately clear that the non-regularized ranking policy is a better metric."**
>
> We are not entirely sure we understood your question: what is the “non-regularized ranking policy,” and what would it mean for this policy to be a better metric? Thank you!
>
> **"In most of the cases, the ability of maximizing the reward/ranking accuracy, while maintaining a moderate KL divergence is more suitable to evaluate the algorithm performance in my viewpoint. Therefore, the ranking accuracy alone is also not a good learning target, which is also evidenced by figure 4 of this work."**
>
> We agree that optimizing for ranking accuracy alone is not necessarily a good learning target; to clarify, we are not suggesting that one should be optimizing ranking accuracy. Rather, we use ranking accuracy as a lens to better understand what models learn during DPO training, as well as when it correlates with win rate.
>
> **"However, in [SimPO’s] experiments, as the win rates evaluated on the standard benchmark significantly increase, we also observe a significant drop in some academic benchmarks, particularly those on the mathematical problems solving… I feel that the paper can still be largely improved if a more comprehensive evaluations can be done, like the trade-off between the performance on instruction-following benchmarks (alpaca-eval, mt-bench, gpt4 win rate) and other academic benchmarks"**
>
> If we understand correctly, the reviewer seems to be suggesting that, in addition to assessing win rate on the query distribution of interest, we should evaluate the models in the paper on benchmarks for other domains. We believe this is out of scope of this paper, which focuses on the fact that DPO does not even yield correct rankings on the training data. We do agree broadly with the reviewer’s sentiment, though, that it is useful to assess top-performing models on a wide variety of benchmarks.

---

> ### Comment · Reviewer_uW8U · 2024-08-07
> **Thanks for the responses**
>
> The ``non-regularized ranking policy'' refers to equation (1), which does not consist of the regularization of the reference model in the original DPO reward.
>
> My main point is that I do not think dropping the reference is a good idea. This is because even with the strong KL regularization, the DPO algorithm (and its variants) already suffers from considerable alignment tax. Therefore, chasing for a high ranking policy is too aggressive in my viewpoint, because it typically leads to performance regression in the out-of-distribution tasks. These have been widely observed in almost all the practices with DPO algorithm and has been mentioned in many original technical report of PPO-based framework (e..g instruct-gpt), as well as the more recent kto and simpo [1].
>
>
> [1] The Good, The Bad, and The Greedy: Evaluation of LLMs Should Not Ignore Non-Determinism

---

> > ### Author Response · Authors · 2024-08-07
> > **Thanks for the clarification!**
> >
> > **“My main point is that I do not think dropping the reference is not a good idea.”**
> >
> > Thank you for the clarification! We agree with you on this point, and our paper does not promote dropping $\pi_\text{Ref}$ altogether without other interventions to prevent generation degradation. (SimPO, for example, uses a target reward margin, on-policy training data, and length normalization *in addition* to removing $\pi_\text{Ref}$.) Our experiments with $L_\text{DPO}^\gamma$ in Section 5 also explore the concept of modifying the strength of the $\pi_\text{Ref}$ terms rather than completely removing them. We show in Figure 10b that some values of $\gamma$ offer an improved tradeoff of ranking accuracy vs. win rate when compared to DPO (DPO results are in Figure 10a, and $L_\text{DPO}^\gamma$ results are in Figure 10b).

---

### Official Review · Reviewer_5qJ8 · 2024-07-09

**Soundness:** 4
**Presentation:** 3
**Contribution:** 3
**Rating:** 6
**Confidence:** 4

**Summary:**

The paper investigates the effectiveness of preference learning algorithms, RLHF and DPO, in training LLMs to rank human-preferred outputs higher. It challenges the assumption that these algorithms can successfully teach models to rank outputs according to human preferences. Empirical findings show that most state-of-the-art models achieve low ranking accuracy, often below 60% on standard datasets. The paper identifies a significant alignment gap due to the limitations of the DPO objective and provides a theoretical analysis of the difficulty in correcting ranking errors. It concludes by highlighting the need for a reevaluation of these algorithms and calls for more nuanced analyses of preference training dynamics.

**Strengths:**

1. The paper provides a thorough empirical analysis, revealing the limited effectiveness of RLHF and DPO in improving ranking accuracy, which is a significant contribution to the field.

2. It offers theoretical insights into why these algorithms fail to correct ranking errors, providing a deeper understanding of the underlying issues in preference learning for LLMs.

3. The findings have clear practical implications for the development of alignment techniques in LLMs, motivating the need for more effective methods to train models that better reflect human preferences.

**Weaknesses:**

1. While the paper suggests a link between preference learning and win rate, it would be strengthened by a more explicit exploration of how these metrics interrelate. Specifically, the authors could benefit from a deeper analysis of the implications of using ranking accuracy versus win rate as measures of model performance, including potential discrepancies and how they might be reconciled.

2. The paper identifies a significant determinism of ranking accuracy by reference model likelihoods, which is a crucial observation. However, it would be advantageous for the authors to expand on this point by discussing potential techniques or algorithmic adjustments that could alleviate the heavy reliance on the reference model. This would bolster the paper's contribution by addressing a key challenge in the field of preference learning for LLMs.

**Questions:**

See Weaknesses.

**Limitations:**

Yes, the authors fully address the limitations and potential social implications of their work.

---

> ### Author Rebuttal · Authors · 2024-08-06
>
> We thank the reviewer for recognizing the “clear implications” of our theoretical and empirical analysis. We answer additional questions below.
>
> **"Specifically, the authors could benefit from a deeper analysis of the implications of using ranking accuracy versus win rate as measures of model performance, including potential discrepancies and how they might be reconciled."**
>
> Thank you for this suggestion. We agree that it would be interesting to further study when and to what extent offline metrics like ranking accuracy are indicative of generative behaviors from the model. Our experiments in Section 5 initiate a study of this, and we find that when the model is close to the reference model, the win rate and the ranking accuracy trend together. We hope that future work can study how different design choices, such as on-policy data (lines 269-275), affect this relationship.
>
> **“The paper identifies a significant determinism of ranking accuracy by reference model likelihoods, which is a crucial observation. It would be advantageous for the authors to expand on this point by discussing potential techniques or algorithmic adjustments that could alleviate the heavy reliance on the reference model.”**
>
> We agree that it is important to find better ways to regularize toward the reference model. Our formulation of the $L_\text{DPO}^\gamma$ objective in Equation 4 is one possibility, and we discuss several others in lines 269-275. We furthermore note that several prior/concurrent works, including [CPO](https://arxiv.org/abs/2401.08417) and [SimPO](https://arxiv.org/abs/2405.14734), remove the reference model entirely and instead modify the objective and dataset to prevent the generative behavior from degrading during preference learning.

---

> > ### Comment · Reviewer_5qJ8 · 2024-08-08
> >
> > Thanks for the responses, and I am keeping my overall score at 6.

---

### Official Review · Reviewer_j6Ru · 2024-07-12

**Soundness:** 2
**Presentation:** 2
**Contribution:** 2
**Rating:** 3
**Confidence:** 4

**Summary:**

This paper studies the preference learning algorithms like DPO in Large Language Models (LLMs). It focuses on the ranking accuracy and shows that existing algorithms fail to achieve high ranking accuracy. Then, the paper further proposes an idealized ranking accuracy and finds that there is an alignment gap.

**Strengths:**

- A metric called ranking accuracy and a variant of this metric are proposed.

- Extensive empirical observations of DPO are provided, which may be interesting for algorithm analysis and understanding.

**Weaknesses:**

- The motivation of studying the rank accuracy is weak and not justified well.

- The analysis and results are somewhat superficial.

**Questions:**

The contributions, compared with similar works, are not clear for reviewers to give an acceptance recommendation.

[1] Xu, Shusheng, et al. "Is DPO superior to PPO for LLM alignment? A comprehensive study." arXiv preprint arXiv:2404.10719 (2024).

[2] Tajwar, Fahim, et al. "Preference fine-tuning of LLMs should leverage suboptimal, on-policy data." arXiv preprint arXiv:2404.14367 (2024).

[3] Tang, Yunhao, et al. "Understanding the performance gap between online and offline alignment algorithms." arXiv preprint arXiv:2405.08448 (2024).

The analysis of PPO in terms of ranking accuracy is missing, which somehow makes the conclusion regarding on-policy and off-policy weak.

**Limitations:**

- The motivation for studying ranking accuracy is weak. The ranking accuracy, especially when defined on the offline dataset, usually does not correlate with the quality of generation behaviors. Let us split the response set into two parts, R_good and R_bad. In practice, we care that sum\_{y} P( y in R_good) > sum\_{y'} P(y' in R_bad). The proposed ranking accuracy, instead cares about P(y in R_good) > P(y' in R_bad), where y and y' are two events from R_good and R_bad, respectively.  Thus, a good generative model can provide many good responses with high probability and, at the same time, it may have a low ranking accuracy. This says that ranking accuracy measures the worst-case behavior of generative models. Thus, we do not need to bother ourselves to care about the ranking accuracy, especially for a small dataset where only a few pairs of preference are given.
- The analysis in terms of ranking accuracy is somehow superficial. In fact, it is the algorithmic bias that DPO introduces the KL-regularized optimization, and as such, it cannot recover the preference-generating distribution. Briefly, the BTL assumption does not invovle the KL term. See the recent work in [1] for more discussion (since this work was posted after NeurIPS submission deadline, it does not affect the review decision). This can also be reflected in equation (2): assume that pi_REF is a uniform policy so its effect can be eliminated and the parameter beta = 1, we see that it recovers the preference-generating policy in terms of alpha. The BTL model assumes that alpha + 1 - alpha = 1, however it may not be true that pi*(y\_w|x) + pi*(y\_l|x) = 1.  The procedure I mentioned is also reflected in equation (4), where the authors introduce a new loss. It in fact corresponds to add a entropy regularization to eliminate the bias of pi\_ref.

Based on the considerations mentioned above, the reviewer does not clearly perceive the technical depth and the contributions of this paper.

[1] Xiao, Jiancong, et al. "On the Algorithmic Bias of Aligning Large Language Models with RLHF: Preference Collapse and Matching Regularization." *arXiv preprint arXiv:2405.16455* (2024).

---

> ### Author Rebuttal · Authors · 2024-08-06
>
> Thank you to the reviewer for your thoughtful comments and feedback. We provide responses to your concerns below:
>
> **"The motivation for studying ranking accuracy is weak.”**
>
> We study ranking accuracy because DPO directly optimizes for ranking accuracy. For example, the authors of the DPO paper state that "the DPO update increases the relative log probability of preferred to dispreferred response," which corresponds exactly to increasing the ranking accuracy. Our Corollary 3.3 + idealized ranking accuracy results (Table 1) also formalize this. Nevertheless, our paper shows that DPO fails to improve ranking accuracy even though it induces desirable behaviors such as win rate improvement, motivating the need for further investigation. Our analyses help us understand better what the model actually learns during DPO training (if not ranking), and can inform future improvements to preference learning algorithms.
>
> **"The analysis of PPO in terms of ranking accuracy is missing, which somehow makes the conclusion regarding on-policy and off-policy weak."**
>
> Figure 1b and Table 1 contain the ranking accuracies of multiple RLHF-PPO models. Theorem 3.1 and Corollary 3.3 also apply to both RLHF-PPO and DPO.
>
>
> **“The ranking accuracy, especially when defined on the offline dataset, usually does not correlate with the quality of generation behaviors."**
>
> We agree, and Section 5 goes further to plot these correlations explicitly. We find that when the model is close to the reference model, the win rate and the ranking accuracy trend together, but as training goes on, they become anti-correlated. **Our paper does not argue that ranking accuracy should replace the win rate metric.** Instead, we demonstrate that DPO does not accomplish what it was designed to do (*i.e.* optimize ranking accuracy), despite its success at improving win rate.
>
> **"Let us split the response set into two parts, R_good and R_bad. In practice, we care that sum_{y} P( y in R_good) > sum_{y'} P(y' in R_bad)... This says that ranking accuracy measures the worst-case behavior of generative models. Thus, we do not need to bother ourselves to care about the ranking accuracy"**
>
> The case brought up by the reviewer concerns generalization, while our analysis is concerned with whether DPO even achieves its own training objective, i.e., whether the model achieves high ranking accuracy **on its own training dataset**. This is a much easier case.
>
> **"The analysis in terms of ranking accuracy is somehow superficial. In fact, it is the algorithmic bias that DPO introduces the KL-regularized optimization, and as such, it cannot recover the preference-generating distribution."**
>
> We respectfully disagree. We agree with the reviewer that even the target distribution of DPO and RLHF-PPO may not have perfect ranking accuracy even on the training data (see discussion on idealized models in Section 3.2, and note that the Table 1 results under $\mathcal{R}^*$ are often less than 100%). However, our results show that there is a gap even between the idealized target distributions and models in practice (see Table 1). In other words, the reviewer’s analysis is not sufficient to explain the results of models in practice; to fill in this gap, we investigate what happens throughout training (Section 4 and 5).
>
> Xiao et al. provides an interesting and complementary perspective on our work, but their proposed regularizer appears to promote higher perplexity in general (see [Table 2 in Xiao et al.](https://arxiv.org/pdf/2405.16455v1#page=24.44)). Rather than resolving the "algorithmic bias" described by Xiao et al., we investigate the gap between on- and off-policy behavior.

---

### Official Review · Reviewer_yhyh · 2024-07-14

**Soundness:** 3
**Presentation:** 3
**Contribution:** 3
**Rating:** 5
**Confidence:** 3

**Summary:**

The paper empirically highlights a few potential flaws in RLHF and DPO that prevent preference-tuned models from achieving high ranking accuracy. It presents a collection of empirical and theoretical findings:
+ Existing preference-tuned models achieve low ranking accuracies.
+ The idealized policy and the DPO/RLHF policy has a significant gap in their ranking accuracy.
+ Preference learning algorithms such as DPO rarely flip incorrect rankings throughout the training process.
+ Ranking accuracy and win rate are positively correlated when models are close to the reference model.

**Strengths:**

The paper provides a thorough empirical and theoretical examination of the performance of preference learning algorithms, particularly RLHF and DPO, in aligning LLMs with human preferences. The paper highlights the significant alignment gap between the observed and idealized ranking accuracies.
The empirical study is extensive, covering a variety of state-of-the-art preference-tuned models across multiple datasets, providing robust evidence supporting its claims on the gap between the idealized and learnt policy under the DPO/RLHF objective.

**Weaknesses:**

1. While it is already argued in the paper that ranking accuracy is practically more meaningful than the reward accuracy, it is still questionable whether totally discarding $\pi_{ref}$ is reasonable. Specifically, the paper should have included experiment results on whether there is also a gap in reward accuracy between the learnt and ideal policy. After all, the DPO and RLHF algorithm explicitely maintain the reference model $\pi_{ref}$, and the implicit reward should be indicative on human preference.

2. Due to the small size of preference labels for any given response pair (1 for most dataset, and 4 for the Alpaca Farm as stated in Sec C.4), it is more reasonable to consider the soft ranking accuracy instead of the hard ranking accuracy. The authors could have reported both or discuss on whether hard ranking accuracy is more meaningful.

**Questions:**

1. One concern is that the ranking accuracy considered here is the hard accuracy (accuracy of the Bayesian optimal classifier). I wonder if the "soft" ranking accuracy will exhibit more noticable improvement after preference fine-tuning. The soft ranking accuracy is $\frac{\pi(y_w)}{\pi(y_w) + \pi(y_l)}$ instead of hard thresholding the predicted accuracy.

2. It is not clear to me what is the main failing part in DPO/RLHF training for high ranking accuracy. In Theorem 3.1, it seems that the perfect RLHF policy should encode the information from $\pi_{ref}$ and the dataset perfectly at the same time. Does the DPO/RLHF model fail to capture $\pi_{ref}$ or the preference dataset?
Have the authors tried to plot the quantity $\frac{\pi_{DPO}(y_w)\pi_{ref}(y_l)}{\pi_{DPO}(y_w)\pi_{ref}(y_l)}$ and compare it to $(\frac{\alpha}{1-\alpha})^{\beta}$ (perhaps a scatter plot)? This can illustrate more clearly what DPO/RLHF failed in reaching the perfect policy.

---

> ### Author Rebuttal · Authors · 2024-08-06
>
> We thank the reviewer for their valuable feedback and suggestions. We appreciate your recognition of the thoroughness of our work. We have included our responses to your questions below and new results in the PDF attached to the global rebuttal.
>
> **"One concern is that the ranking accuracy considered here is the hard accuracy (accuracy of the Bayesian optimal classifier). I wonder if the "soft" ranking accuracy will exhibit more noticable improvement after preference fine-tuning"**
>
> Thank you for this suggestion – we provide plots of the soft ranking accuracy during training for GPT2, Pythia 2.8B, and Llama 2 7B in Fig. 1 of the PDF attached to the global rebuttal. While there is some gradual improvement for the "incorrect->correct" group, there is little to no improvement for the examples that stay incorrectly ranked (i.e. "incorrect->incorrect" group).
>
> **"While it is already argued in the paper that ranking accuracy is practically more meaningful than the reward accuracy, it is still questionable whether totally discarding $\pi_\text{Ref}$ is reasonable. Specifically, the paper should have included experiment results on whether there is also a gap in reward accuracy between the learnt and ideal policy."**
>
> The reason we consider ranking accuracy instead of reward accuracy is, as mentioned in lines 133-134, "we ultimately sample from $\pi_\theta$ rather than $\frac{\pi_\theta}{\pi_\text{ref}}$." Moreover, the reward accuracy of many models can be quite high even if the models have terrible ranking accuracy. For example, consider an example $(x,y_w,y_l)$ where $\pi_\text{Ref}(y_w|x)=0.1,\pi_\text{Ref}(y_l|x)=0.8,\pi_\theta(y_w|x)=0.11,\pi_\theta(y_l|x)=0.72$. Then the ranking accuracy is 0 on this example because $\pi_\theta(y_w|x)=0.11<0.72=\pi_\theta(y_l|x)$ but the reward accuracy is 1 because $\log(\pi_\theta(y_w|x)/\pi_\text{Ref}(y_w|x))=\log(1.1)>\log(0.9)=\log(\pi_\theta(y_l|x)/\pi_\text{Ref}(y_l|x))$. Thus, reward accuracy is a much less useful metric to consider.
>
> **"It is not clear to me what is the main failing part in DPO/RLHF training for high ranking accuracy. In Theorem 3.1, it seems that the perfect RLHF policy should encode the information from $\pi_\text{Ref}$ and the dataset perfectly at the same time. Does the DPO/RLHF model fail to capture $\pi_\text{Ref}$ or the preference dataset?”**
>
> First, we would like to clarify that it is not always possible to encode both the reference model and the dataset perfectly, since there may be conflicts between the two. Indeed, the takeaway from Theorem 3.1 is that the optimal policy encodes the dataset preferences scaled by the reference model, so it is not always possible for the model to recover the true preference probability $\alpha$ if $\beta$ is not set correctly (see Table 1). We will be sure to clarify this in our next revision.
>
> Section 4 studies why models fail to achieve high ranking accuracy in practice. Section 4.2 in particular demonstrates that when the reference model has an incorrect ranking on a datapoint, the DPO loss must be reduced to a very small value to correct the ranking (Theorem 4.1 and Figure 3). This value is so small that DPO typically overfits before it reduces the loss to this value (as demonstrated by Figures 2a and 3.) Thus, in practice, it is difficult to optimize the model to achieve the idealized ranking accuracy described in Section 3.
>
> **"Have the authors tried to plot the quantity $\frac{\pi_\text{DPO}(y_w)\pi_\text{ref}(y_l)}{\pi_\text{DPO}(y_w)\pi_\text{ref}(y_l)}$ and compare it to $(\frac{\alpha}{1−\alpha})^\beta$ (perhaps a scatter plot)? This can illustrate more clearly what DPO/RLHF failed in reaching the perfect policy."**
>
> Thank you for this suggestion. We agree that if we had access to the true proportions of raters who preferred one response over the other, denoted $\alpha$, this analysis would be insightful. Unfortunately, most preference training datasets report only binary preference values, as we discuss in lines 65-76. However, our analyses in Figures 2 and 3 provide a clear diagnosis of the failing – DPO encourages the model to continually increase the reward margins, and flipping the actual ranking is difficult (Theorem 4.1).

---

> > ### Comment · Reviewer_yhyh · 2024-08-09
> >
> > I appreciate the authors' detailed responses and have reviewed both their individual and global responses. I maintain my current stance and recommend acceptance.

---

### Author Rebuttal · Authors · 2024-08-06

We are grateful to the reviewers for their detailed feedback – Reviewer yhyh notes the paper's "robust evidence supporting its claims," Reviewer 5qJ8 comments that our "findings have clear practical implications for the development of alignment techniques," and Reviewer uW8U notes "the authors identify an interesting question and the present a complete story with solid evidence in both theory and experiments…overall, I support the acceptance."

On the other hand, the reviewers expressed some confusion about the choice to analyze ranking accuracy. We focus on this metric not because we believe it to be the best metric of model performance, but because algorithms like DPO *directly optimize for ranking accuracy*. We show that since DPO does not successfully optimize its objective *even on the training dataset* (but does meaningfully improve generation quality as measured by win rate), the learning mechanisms behind such preference learning algorithms deserve further attention. We will be sure to clarify this point in future revisions.

We additionally attach a PDF containing the soft ranking accuracy results requested by Reviewer yhyh.

---

### Decision · Program_Chairs · 2024-09-25

**Decision:**

Accept (poster)

**Comment:**

This paper studies preference learning with RLHF and DPO for LLMs.  It highlights some enlightening empirical findings, such as the existing preference learning algorithms obtain low ranking accuracies, the idealized policy and these methods' policies have a significant ranking accuracy gap.  They show that when the model is close to reference the win rate and preference accuracy are strongly correlated.  The reviewers praise the extensive empirical comparisons and how well their claims are supported by evidence.  They also praise the theoretical insights that can explain where these method fail.  The authors clearly rebut criticisms such as why they prefer ranking accuracy over reward accuracy, and they provide results for soft-ranking accuracy in addition to the hard ranking accuracy.